# Low Tensor Rank Learning of Neural Dynamics

**Arthur Pellegrino**
School of Informatics
University of Edinburgh
`pellegrino.arthur@ed.ac.uk`

**N Alex Cayco-Gajic**[*]
Département d'Etudes Cognitives
Ecole Normale Supérieure
`natasha.cayco.gajic@ens.fr`

**Angus Chadwick**[*]
School of Informatics
University of Edinburgh
`angus.chadwick@ed.ac.uk`

## Abstract

Learning relies on coordinated synaptic changes in recurrently connected populations of neurons. Therefore, understanding the collective evolution of synaptic connectivity over learning is a key challenge in neuroscience and machine learning. In particular, recent work has shown that the weight matrices of task-trained RNNs are typically low rank, but how this low rank structure unfolds over learning is unknown. To address this, we investigate the rank of the 3-tensor formed by the weight matrices throughout learning. By fitting RNNs of varying rank to large-scale neural recordings during a motor learning task, we find that the inferred weights are low-tensor-rank and therefore evolve over a fixed low-dimensional subspace throughout the entire course of learning. We next validate the observation of low-tensor-rank learning on an RNN trained to solve the same task. Finally, we present a set of mathematical results bounding the matrix and tensor ranks of gradient descent learning dynamics which show that low-tensor-rank weights emerge naturally in RNNs trained to solve low-dimensional tasks. Taken together, our findings provide insight on the evolution of population connectivity over learning in both biological and artificial neural networks, and enable reverse engineering of learning-induced changes in recurrent dynamics from large-scale neural recordings.

## 1 Introduction

Populations of neurons perform tasks through their dynamics, and these computations can be understood through the lens of recurrent neural networks (RNNs) [1, 2]. Recent work has shown that RNNs trained on idealized versions of behavioural tasks from neuroscience experiments can be reverse-engineered to better understand the dynamical principles by which they perform tasks [3, 4, 5, 6, 7]. RNNs can also be fitted to neural data to infer the latent dynamics that drive neural activity in specific tasks [8, 9, 10, 11]. However, an understanding of task learning in the brain requires methods to understand how these latent dynamics evolve over trials, and to map these computational changes to learning-induced changes in neural connectivity [12, 13]. For example, it has been observed that gradient descent in task-trained RNNs tends to drive low-rank weight updates [14]. Yet the structure of learning dynamics itself remains largely unknown, both in the context of gradient-based optimization of neural networks and biological learning in neural systems.

To address this question, we consider how RNN dynamics evolve as a result of structured changes in connectivity over learning. Specifically, we consider changes in RNN connectivity over multiple

---

[*]These last authors contributed equally.

37th Conference on Neural Information Processing Systems (NeurIPS 2023).

trials by assuming that the weight tensor, i.e. the 3-tensor formed by stacking the weight matrix over all trials, has low tensor rank. This allows us to identify how distinct components of the weight matrix vary over trials while simultaneously restricting the parameter complexity and benefiting from the interpretable framework of low (matrix) rank RNNs at each trial [15]. Furthermore, by imposing a constraint on the covariance of the trial factors we are also able to separate smooth changes over learning from condition-specific variability on individual trials. In contrast to classic tensor decomposition methods (such as PARAFAC), which instead constrain the neural activity itself to be low rank, *low-tensor-rank RNNs* (ltrRNNs) capture high-dimensional neural activity resulting from nonlinear neural dynamics, while preserving the interpretability of these linear methods through having interpretable low-dimensional factors.

**Main contributions.** We first apply this method to neural data during a motor learning task [16] to show that the resulting neural activity can be captured with surprisingly low-tensor-rank weight updates. Next, we aim to find intuition for the observation that learning can be low tensor rank by turning to gradient-based optimization, which has recently been shown to be able to explain changes in neural activity patterns over motor learning [17, 18]. Towards this end, we show numerically that an RNN that is trained to perform the same task also results in low-tensor-rank learning dynamics. Finally, we provide analytical intuition in the form of upper bounds on both the matrix rank and the tensor rank of gradient-based optimization in RNNs. Ultimately these results provide evidence for low tensor rank learning structure of neural dynamics both in the brain and in RNNs.

## 2  Low tensor rank learning

**The weight tensor.** Learning in recurrent neural networks, both biological and artificial, involves the continual update of a weight matrix $W \in \mathbb{R}^{N \times N}$, where each element $W_{ij}$ represents the connectivity from neuron $j$ to $i$. Here, we consider a set of $K$ discrete samples of the weight matrix over learning: $W^{(k+1)} = W^{(k)} + \Delta W^{(k)}$ for $k \in \{0, ..., K-1\}$. In neuroscience, $\Delta W^{(k)}$ could represent plasticity-induced changes in connectivity strength following each trial in the experiment (e.g., a perceptual decision or motor action), whereas in machine learning they could represent weight updates arising from the application of a given learning rule to a set of training data. Learning can then be summarised by the *weight tensor* $\mathbf{W} \in \mathbb{R}^{N \times N \times K}$ formed by stacking together the weight matrices $W^{(k)}$ on all trials[1], i.e., $\mathbf{W}_{ijk} = W_{ij}^{(k)}$.

Here, we investigate the multilinear structure of the weight tensor $\mathbf{W}$ in order to gain insight into the relationship between learning, recurrent weights, and network dynamics (Fig. 1a). In particular, given recent studies suggesting that trained RNNs are typically low (matrix) rank [14], we ask whether the weight tensor is *low tensor rank*, i.e., if the weight tensor can be written as a sum of $R$ rank-1 components:

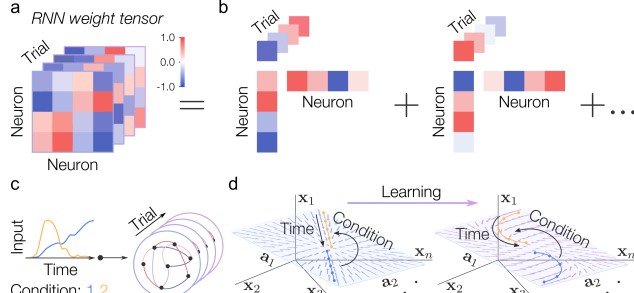

$$\mathbf{W} = \sum_{r=1}^{R} \mathbf{a}_r \otimes \mathbf{b}_r \otimes \mathbf{c}_r$$

$$W_{ij}^{(k)} = \sum_{r=1}^{R} \underbrace{c_r^{(k)}(\mathbf{a}_r \otimes \mathbf{b}_r)_{ij}}_{=a_{r,i}b_{r,j}c_r^{(k)}}$$

where $\mathbf{a}_r, \mathbf{b}_r \in \mathbb{R}^N$ and $\mathbf{c}_r \in \mathbb{R}^K$ for $r \in \{1, \dots, R\}$ (Fig. 1b). A conse-

Figure 1: **Low tensor rank recurrent neural networks**. **a.** We consider the 3-tensor formed by stacking the weights over learning. **b.** Constraining the weight tensor to be low rank allows for an interpretable analysis of the evolution of the weights over learning. **c.** LtrRNNs partition neural variability into task condition-specific inputs and learning-related weight changes over trials. **d.** LtrRNNs capture changes in dynamics constrained to a low-dimensional subspace, which reshape neural representations over learning.

quence of the low tensor rank assumption is that the weights $\text{vec}(W^{(k)})$ must evolve within an $R$-dimensional subspace of $\mathbb{R}^{N^2}$ spanned by the set of rank-1 matrices $\{\text{vec}(\mathbf{a}_r \otimes \mathbf{b}_r)\}_{r=1}^{R}$.

---

[1]For consistency with our analysis of neural data, we use the terminology *trial* throughout to describe the index of the weight samples $k$, denoted in superscript (cf. neuron and time indices denoted in subscript).

**Low tensor rank RNNs.** To investigate the structure of weight tensors formed over learning in recurrent networks, we consider a continuous-time RNN (Fig. 1c). The RNN dynamics on trial $k$ are given by:

$$\tau\dot{\mathbf{x}}^{(k)} = W^{(k)}\phi(\mathbf{x}^{(k)}) - \mathbf{x}^{(k)} + B\mathbf{u}^{(k)}(t)$$

for $B \in \mathbb{R}^{N \times N_{\text{input}}}$, $\mathbf{u}^{(k)}(t) \in \mathbb{R}^{N_{\text{input}}}$. If the low tensor rank hypothesis is satisfied, the system can be written as:

$$\tau\dot{\mathbf{x}}^{(k)} = \sum_{r=1}^{R} c_r^{(k)}(\mathbf{a}_r \otimes \mathbf{b}_r)\phi(\mathbf{x}^{(k)}) - \mathbf{x}^{(k)} + B\mathbf{u}^{(k)}(t),$$

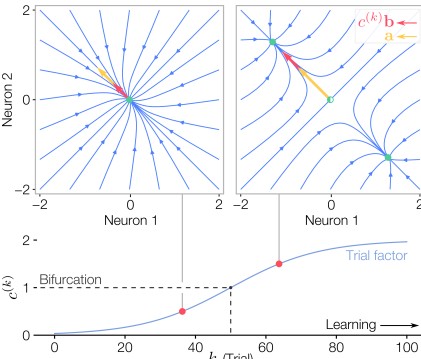

which reduces to a low-rank RNN on every trial [15]. As a result, $\mathbf{x}^{(k)}(t)$ is constrained to an $(R + N_{\text{input}})$-dimensional subspace of $\mathbb{R}^N$ spanned by $\{\mathbf{a}_r\}_{r=1}^{R} \cup \{B_i\}_{i=1}^{N_{\text{input}}}$, and this subspace is fixed across trials (Fig. 1d).

The weight matrix evolves across trials via a simple rescaling of the rank-1 components by the trial factors $c_r^{(k)}$. This may at first appear to restrict the possible changes in dynamics over trials to simple scalings of the flow field along the directions determined by the corresponding $\mathbf{a}_r \otimes \mathbf{b}_r$ components of the weights. In fact, non-trivial changes in the flow field (e.g., bifurcations) can occur due to the nonlinear activation $\phi(\mathbf{x})$ (Fig. 2; Supplementary Material A). Moreover, even in the case of a linear ltrRNN ($\phi$ =id), a rich repertoire of dynamics is possible. First, the system can switch between different vector fields by allowing the corresponding trial factors to transition to zero (or nonzero) values. More generally, the eigenvalues are non-trivially related to the $\mathbf{c}_r$'s. The real eigenvalues can be shown to satisfy the following equality (Supplementary Material A):

Figure 2: **Bifurcation in the flow field of an ltrRNN**. Here $\mathbf{a} = \mathbf{b} = [-1/\sqrt{2}, 1/\sqrt{2}]^T$, so that the weights at any given trial $k$ are, $W^{(k)} = \frac{c^{(k)}}{2}\begin{bmatrix} 1 & -1 \\ -1 & 1 \end{bmatrix}$. A supercritical pitchfork bifurcation occurs at $c^{(k)} = 1$ when $\phi = \tanh$; for $\phi$ =id (Sup. Fig. 1) a line attractor emerges at $c^{(k)} = 1$.

$\lambda_r = \frac{1}{2}\cos(\theta_{rr}^{\tilde{\mathbf{v}},\mathbf{v}})^{-1}\sum_{r'} c_{r'}[\cos(\theta_{rr'}^{\tilde{\mathbf{v}},\mathbf{a}} + \theta_{rr'}^{\mathbf{v},\mathbf{b}}) + \cos(\theta_{rr'}^{\tilde{\mathbf{v}},\mathbf{a}} - \theta_{rr'}^{\mathbf{v},\mathbf{b}})]$, where $\tilde{\mathbf{v}}_r$ and $\mathbf{v}_r$ are the $r$th left and right eigenvectors and $\theta_{rr'}^{\mathbf{x},\mathbf{y}}$ is the angle between the $r$th $\mathbf{x}$ and $r'$th $\mathbf{y}$. This non-trivial relationship between the eigendecomposition and tensor rank decomposition endows considerable flexibility in terms of the possible changes in dynamics over trials, depending on the relative orientations of $\{\mathbf{a}_{r'}\}_{r'=1}^{R}$ and $\{\mathbf{b}_{r'}\}_{r'=1}^{R}$ as well as the angle $\theta_{rr}^{\tilde{\mathbf{v}},\mathbf{v}}$ which quantifies the non-normality of the resulting weight matrix.

**Fitting ltrRNNs to data.** These results suggest that ltrRNNs comprise a highly flexible and expressive class of neural networks for relating the changes in recurrent dynamics to low-dimensional structure of the weight updates. We next considered how ltrRNNs could be fit to a dataset (for example, the activations of an RNN over the course of training, or recordings from a population of neurons as an animal learns to perform a task) in order to elicit a low tensor rank description of the weight updates governing the evolution of the dynamics over trials.

We consider a dataset in the form of a tensor $\mathbf{Y} \in \mathbb{R}^{N_{\text{data}} \times T \times K}$, comprising the activity of $N_{\text{data}}$ neurons measured on $K$ trials, each of duration $T$ time points, with single trial activations $\mathbf{y}^{(k)}(t)$. To fit the ltrRNN to such a tensor, we minimise the loss function $L(\mathbf{a}, \mathbf{b}, \mathbf{c}, B, M) = \sum_{t,k}^{T,K}\|M\phi(\mathbf{x}^{(k)}(t)) - \mathbf{y}^{(k)}(t)\|^2$, where $M \in \mathbb{R}^{N_{\text{data}} \times N}$ is a readout matrix mapping the activation of the ltrRNN units onto the neural data. The loss function is minimised directly via gradient descent with respect to the parameters of a rank $R$ tensor decomposition of $\mathbf{W}$.

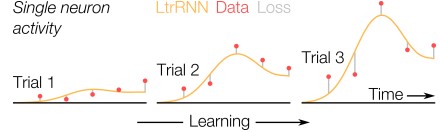

Figure 3: **LtrRNN fit to neural data**. LtrRNNs can be fitted to neural population data (here shown for a single neuron and for three trials), thus capturing smooth changes in activity over learning.

Above we considered how the low tensor rank assumption constrains the possible changes in RNN dynamics for the case in which the $\mathbf{c}_r$ can change arbitrarily across trials. However, here we are interested specifically in understanding how these dynamics change over the course of *learning* which

typically imposes additional structure on the $\mathbf{c}_r$'s. Here, we add an inductive bias that the components determining the weights change smoothly over trials, which we impose by parameterizing the trial factors as $\mathbf{c}_r = (L + \sigma I)\bar{\mathbf{c}}_r$ where $LL^T = A$ is the Cholesky decomposition of the smooth covariance matrix $A$ (assuming $\bar{\mathbf{c}}_r \sim \mathcal{N}(0, I)$, see Supplementary Material A). In contrast, we constrain the inputs to the RNN $\mathbf{u}^{(k)}(t)$ to depend only on *task condition* (i.e., the type of trial, such as the stimulus presented or set of instructions given), thus assuming that most of the condition-independent inter-trial variability in the data comes from learning-induced changes in the dynamics of the system rather than changes in its inputs. By partitioning the variability in this way, we ensure that the activations $\mathbf{x}^{(k)}(t)$, as well as the weights $\text{vec}(W^{(k)})$, evolve in low-dimensional subspaces of $\mathbb{R}^N$ and $\mathbb{R}^{N^2}$, respectively, throughout the course of learning. In the following sections we ask how accurate this low-dimensional view of learning is in the context of both biological and gradient-based learning.

## 3   Related work

LtrRNNs integrate a broad range of concepts in neuroscience and machine learning, including low (matrix) rank RNNs, tensor decompositions, fitting RNNs to neural data, and analytical studies of gradient descent dynamics. Here we review the relationship between ltrRNNs and previous results across these domains.

**Inferring weights from neural activity.** A diverse range of methods have historically been used to infer synaptic connectivity from neural recordings [19]. Our work specifically builds upon a recent line of investigation attempting to infer the recurrently generated dynamics $\dot{\mathbf{x}} = f(\mathbf{x}, W)$ of the network from samples of its trajectories [20]. [21] and [22] inferred learning-induced changes in population dynamics from neural data, but this approach was limited to pre-post comparisons at two timepoints in learning. Others have developed methods to infer the learning rules governing weight updates from post-learning neural activity [23], or spike train recordings [24, 25]. However, no previous study has directly inferred the evolution of weights over the course of learning in neural data. Our low-tensor-rank approach enables this inference by constraining the parameter complexity, which allows more efficient application to neural data. Moreover, in contrast to previous approaches [21, 22] our method ensures that the dynamics at different phases of learning remain jointly interpretable due to the existence of a stable latent space in which the network activity unfolds.

**Low rank RNNs.** Previous work has introduced low rank RNNs as a powerful framework to uncover the low-dimensional dynamics underlying task performance [15, 7], and which can be inferred from neural population data [10]. In the context of learning, one could naively fit a separate low rank RNN to data recorded on each trial, but there is no guarantee that the resulting $\mathcal{O}(NK)$ parameters could be related to each other. By instead assuming the weights have low tensor rank structure, we ensure that smooth changes in the dynamics can be mapped over trials with $\mathcal{O}(N + K)$ parameters, while benefiting from the low rank RNN framework on each trial.

**The dynamics of learning.** Recent work has investigated the dynamics of learning via gradient descent in artificial neural networks [26, 14]. [26] showed that deep linear networks progressively learn the leading singular values of the input-output covariance matrix, which naturally leads to low rank weight matrices when the input-output mapping is itself low rank. [14] extended these findings to show that the weight updates of an RNN are low rank. However, the proof assumed an infinite-dimensional linear dynamical system at steady-state and with Gaussian-distributed weights. Using adjoint sensitivity analysis, our work extends this result by deriving bounds on the matrix rank on the weight updates of finite-dimensional nonlinear networks away from steady-state. Furthermore, to investigate the evolution of weights over learning, we provide bounds on the tensor rank of learning.

**Tensor decomposition methods.** Our framework can be used to infer low dimensional latent structure in neural data. An alternative approach is to directly fit a low rank approximation of the *neuron × time × trial* data tensor [27, 28, 29, 30]. Such linear methods have become increasingly popular because of their interpretability but have two key shortcomings: first, neural representations are generally nonlinearly embedded due to their dynamics and task structure [31, 32, 33]. Second, these methods do not provide insight into the dynamics underlying neural computations. Here, we address both issues by parameterizing the dynamics themselves as being low tensor rank, which allows changes in the dynamics to be mapped directly while also enabling RNN activity over trials to be visualized in a fixed subspace spanned by the weight tensor column factors.

# 4 LtrRNN dynamics capture neural activity during motor learning

To test whether neural population dynamics during learning are consistent with a low tensor rank framework, we fit ltrRNNs of varying rank to recordings from the motor and premotor cortex of the macaque during a motor learning task in which the subject must adapt to a force field perturbation in order to reach the target endpoint [16] (Fig. 4a). Following evidence that motor cortical initial states are set by upstream regions during the preparatory period [16, 34], we parameterize $u^{(k)}(t)$ with a neural ODE (see Supplementary Material B) with solution $\tilde{u}^{(i)}$ on a trial of condition $i$ during motor preparation, after which the dynamics evolved autonomously. As preprocessing, we first Gaussian filtered the spiking activity of each neuron (std $= 40$ ms) to obtain smooth estimates of the instantaneous firing rates. To compare across models, blocks of consecutive entries in time and trials were held-out for cross validation, and the remaining entries of the data tensor were used to fit the parameters of each ltrRNN model. For comparison we also fit a full tensor RNN (i.e. all $N^2 K$ entries of $\mathbf{W}$ were fitted), as well as full and low matrix rank RNNs whose weights stayed constant over all trials. We quantified model performance as the unexplained variance on the interiors of the held out blocks while discarding borders to reduce temporal correlations between the train and test set (Fig. 4b inset, Supplementary Material B).

Interestingly, when we compared the performance of the full tensor RNN to a static RNN, we found only a $\sim 5\%$ difference in the variance explained, indicating that much of the variability in the data is determined by task condition and dynamics, with the difference in performance attributable to learning-induced changes [16]. However, of this remaining variability, a ltrRNN with only 5 components was able to achieve similar performance to the full tensor RNN (Fig. 4b), supporting the hypothesis that learning dynamics are low rank. Using this cross-validation procedure also allowed us to compare the performance of the ltrRNN model with low-rank matrix and tensor decompositions which do not fit the underlying nonlinear neural dynamics. We found that for low ranks, our method outperformed truncated SVD (applied to the trial-concatenated data) and PARAFAC (Fig. 4b). Interestingly, in the case that $\phi =$id, the activity of the ltrRNN is constrained to an $R$-dimensional subspace, therefore the MSE (without cross-validation) is lower-bounded by that of the rank $R$ SVD by the Eckart-Young theorem. This suggests that the the higher performance of ltrRNN compared to PCA is due to the nonlinear mapping from the membrane potential space to the firing rate space.

Since after $t = 0$ the network evolved autonomously, all of the information regarding its trajectory was contained in the the recurrent dynamics and initial state. We find that the initial states inferred by the model reflect the topology of the task variables (Fig. 4c.). In comparison, in the perturbation block of trials there was only a small change to the initial states, consistent with the finding that the majority of the variability in the data was due to changes in task condition rather than learning (Fig. 3b, Supplementary Material B).

Since ltrRNNs reduce to a low-rank RNN on any given trial, their membrane potential $\mathbf{x}(t)$ is constrained to lie in the space spanned by the column vectors $\mathbf{a}_r$ [15]. The membrane potential can therefore be visualized via a projection onto each $\mathbf{a}_r$ to observe the low-dimensional activity of the network [10]. We find that, compared to applying PCA directly on the neural data, ltrRNNs yield more interpretable visualizations of the condition and learning-related variability in the neural (Fig. 3d,e). Additionally, since ltrRNNs parameterize changes in dynamics, we can visualize the vector field in the subspace spanned by the column vectors. Consistent with our task-trained RNN results, and those of the literature [17], small changes in the vector field are sufficient to account for learning the perturbation (Fig. 4f). These changes in the dynamics of the network can be easily interpreted through the trial factors. We find that the dynamics along certain directions in the membrane potential space change during learning (Fig. 4g). Furthermore, consistent with the hand kinematics, and recent experimental evidence [35], some of these changes in the trial factors do not simply revert back to baseline during the washout period (in which the force field perturbation is removed; Fig. 4h,i). Dynamics along some columns capture target variability (Fig. 4j, top 2 components), whereas others capture mainly temporal variability (Fig. 4j, bottom 3 components). Interestingly, the trial factors which revert to baseline during the washout period are those corresponding to target-related dynamics, while those which are persistently changed are the temporal variability factors (Fig. 4i).

Overall, we find that learning-related variability can be accounted for by low-tensor-rank changes in the recurrent dynamics of neural populations. LtrRNN allows uncovering these changes in large-scale neural data and vizualizing their effect in an interpretable fashion.

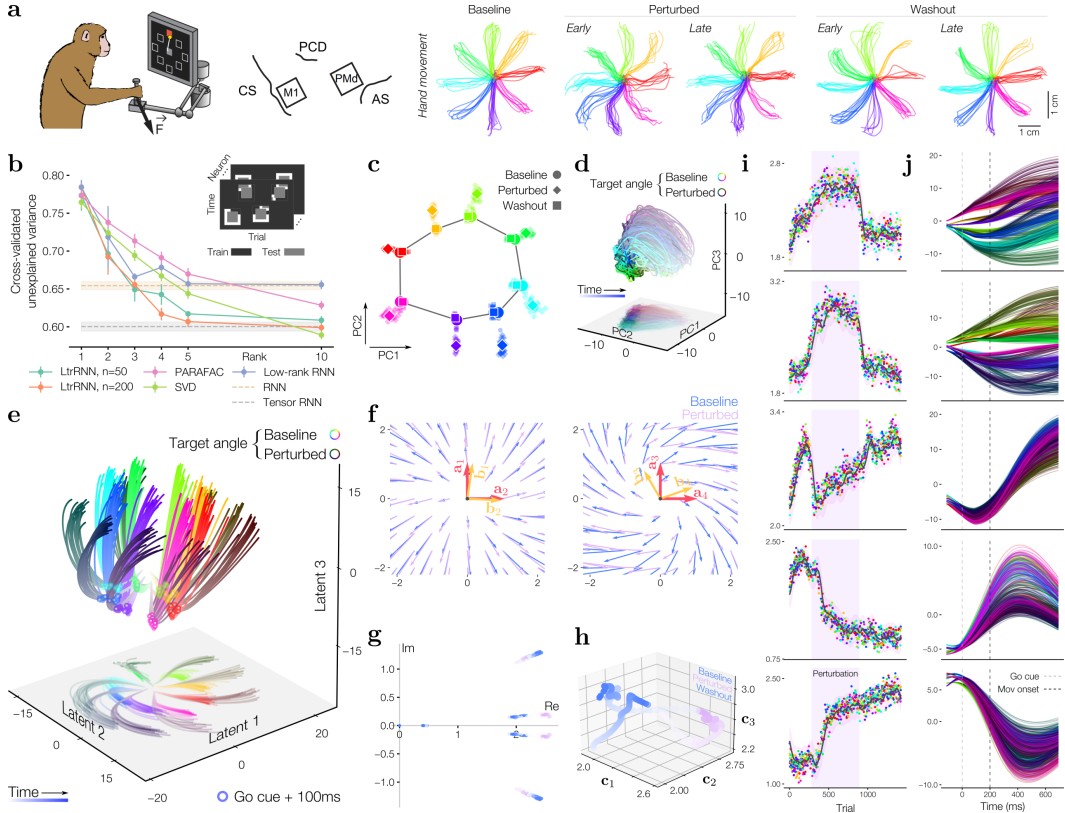

Figure 4: **LtrRNN separates learning- from condition-related variability in motor neural data**. **a.** We apply ltrRNN to recordings of the motor and premotor cortex during learning of a perturbed reach task. (Schematic adapted from [16]). **b**. We hold out for testing blocks of trials and time points (100 ms by 50 trials). This reduces temporal correlations between the train and test sets as compared to holding out individual tensor entries. **c**. State of the ltrRNN ($R = 5$) 100 ms after the go cue. LtrRNN captures the topology of the task. Furthermore, different initial states seem to emerge during the perturbation period. **d**. PCA directly applied to the neural data. **e**. Activity of the ltrRNN projected on the column vectors of the first three components. **f**. Vector field along pairs of column vectors. We constrain the activity to the columns of the tensor such that $\mathbf{x} = q_r \mathbf{a}_r + q_{r'} \mathbf{a}_{r'}$ for $q_r, q_{r'} \in [-2, 2]$ and we compute the vector field $\mathbf{a}_r(\phi(\mathbf{x}) \cdot \mathbf{b}_r) + \mathbf{a}_{r'}(\phi(\mathbf{x}) \cdot \mathbf{b}_{r'})$. **g**. Eigenspectrum of $W^{(k)}$. Saturation gradient indicates trial. **h**. The trial factors $\mathbf{c}_r$ can be seen as a latent variable of learning, such that they describe the evolution of the weights in a low-dimensional subspace of the weight space spanned by $\mathbf{a}_r \otimes \mathbf{b}_r$. **i**. Trial factors. The black line is computed using $\sigma = 0$. **j**. LtrRNN activity projected onto each column factor (ordered as in **i**). In particular the activity along the top two components seem to be more specific to reach angle.

## 5 Low tensor rank learning in task-trained RNNs

We next decided to test the low-tensor-rank learning hypothesis in a model in which we had direct access to the ground truth weight tensor. Towards this end, we trained an RNN with unconstrained weight structure to perform the same motor task.

**Task-trained RNN model**. At any given trial $k$, the RNN linearly controls the force applied to the hand:

$$\ddot{\mathbf{y}}^{(k)} = D\phi(\mathbf{x}^{(k)}(t)) + \boldsymbol{\kappa}^{(k)}(t)$$

where $\boldsymbol{\kappa}^{(k)}(t) \in \mathbb{R}^2$ is an Ornstein-Uhlenbeck process representing noise in the execution of movement [36]. To create a purely ballistic model, we provide the RNN the target position and a hold cue. For the objective we use the integrated hand to target position so that the RNN simply has to push the hand as fast as possible towards the target (Fig. 5a).

After first pre-training the RNN to move the hand to the goal position at the go onset, we then probe motor learning following the same protocol as the neural data. Specifically, the RNN first performs 50 trials in the baseline condition, after which we introduce a force perturbation orthogonal and proportional to the velocity of the hand for 100 trials (Fig. 5b):

$$\ddot{\mathbf{y}}^{(k)} = D\phi(\mathbf{x}^{(k)})(t) + cR\dot{\mathbf{y}}^{(k)}(t) + \boldsymbol{\kappa}^{(k)}(t)$$

where $R \in \mathbb{R}^{2\times 2}$ is a 90° clockwise rotation matrix, and $c \in \mathbb{R}$ is the coefficient of perturbation. Finally, the RNN performs the original unperturbed task for another 100 trials (washout). Throughout the perturbation and washout periods, the weights are updated following SGD.

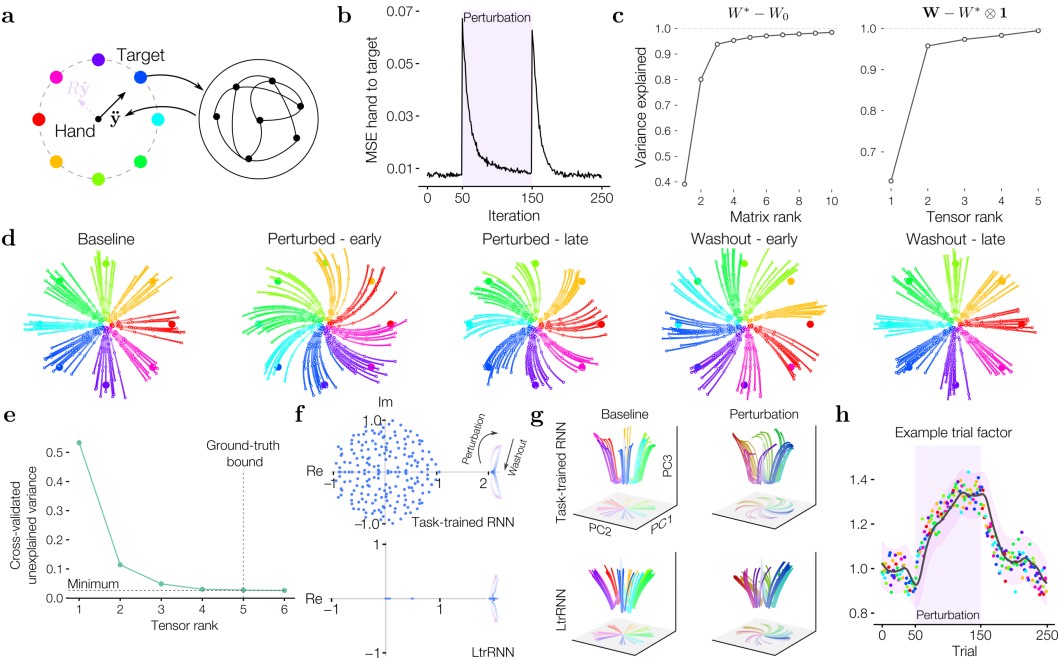

Figure 5: **Learning-induced weight changes in task-trained RNNs are low tensor rank**. **a**. RNN model. **b**. Average MSE between hand and target positions integrated throughout the trial. **c**. Left: Variance explained of the weights resulting from pre-training $W^*$. The original task training results in a rank-3 RNN. Right: Variance explained of the tensor of updates $\mathbf{W} - W^* \otimes \mathbf{1}$ due to retraining. The retraining procedure results in a rank-2 tensor. We further found that the subspaces spanned by the pre-training columns (resp. rows) and retraining columns (resp. rows) were different, suggesting a tensor of rank at most 5. **d**. Hand movements during various periods of learning, where "early" and "late" describe respectively the first and last trial during which the perturbation is introduced or removed. **e**. Using the same cross-validation as in Fig. 4 uncovers the low tensor rank structure. **f**. Eigenspectrum of the weights $W^{(k)}$ over learning. Top: Ground truth weights of the task-trained RNN. Bottom: Weights of the ltrRNN fit to the task-trained RNN activity. Imaginary eigenvalues emerge to counter the rotational effect of the perturbation, and are uncovered by ltrRNN. **g**. Activity projected on PCs. The rotational activity during perturbed trials is uncovered by ltrRNN. **h**. Example of trial factor uncovered by ltrRNN which correlates with learning at the level of the behaviour.

We next analyzed how the structure of the weight matrix changed over the baseline, perturbation, and washout blocks. The change in weights as a result of the pre-training $W^* - W_0$, where $W_0$ is the random initialization, and $W^*$ the weights after pre-training, is of matrix rank 3 (Fig. 5c, left). Since the weight updates resulting from learning the perturbation are small (consistent with the literature [17]), we cannot simply apply PARAFAC on the weights recorded during perturbation learning, as only the 3 rank-1 terms from pre-training will be visible. However, running PARAFAC on the updates $\mathbf{W} - W^* \otimes \mathbf{1}$ (where $\mathbf{1}$ is the vector of ones), reveals tensor rank 2 updates (Fig. 5c, right). Furthermore, the subspace spanned by the columns (or rows) resulting from pretraining is different than those resulting from perturbation learning (Supplementary Material C).

**LtrRNN application**. We next ask whether ltrRNN can uncover this low-tensor-rank structure in the weights, especially the small changes due to perturbation learning. Using our cross-validation procedure, we find that the variance of the activity of the task-trained RNN that is unexplained by the fitted ltrRNN plateaus at tensor rank 5 (Fig. 5e), consistent with our analysis of the ground truth weights. Furthermore, the weights of the ltrRNN share similar spectral properties, and a similar evolution over learning, compared to ground truth (Fig. 5f). In particular, in both cases, imaginary eigenvalues, corresponding to rotational activity, emerge over learning, consistent with the behaviour being rotational post-learning to counter the force field (5d). The emergence of rotational activity is also visible at the level of the neural activity (Fig. 5g).

Therefore, consistent with the results found through ltrRNN fit to neural data, we found that the weight updates in an RNN trained on a perturbed ballistic reach task had low-tensor-rank structure. Furthermore, ltrRNN was able to uncover this structure in the weights of the task-trained RNN, and its evolution over learning.

## 6 Gradient-based learning constrains the tensor rank of weight updates

We have so far observed that learning leads to low-tensor-rank weight changes in both biological data and task-trained RNNs. To gain deeper insight into why this is the case, we next present a set of mathematical results regarding the tensor rank of gradient-based learning in RNNs. Towards this end, we use the method of the adjoint [37, 38] to derive a dynamical system whose solution is the gradient of a loss functional with respect to the RNN weights [39].

**The adjoint.** For a dynamical system $\dot{\mathbf{x}} = f(\mathbf{x}, \boldsymbol{\theta})$, the state adjoint of the loss functional $L : \mathbb{R}^n \to \mathbb{R}$ at a particular time point $t$ is defined as $\mathbf{a_x}(t) = \frac{dL(\mathbf{x}(T))}{d\mathbf{x}(t)}$ where $T$ is the time of loss evaluation (but see Supplementary Material E for the case of a loss functional that is integrated over time). From the dynamics of $\mathbf{x}$, the state adjoint dynamics can be derived:

$$\dot{\mathbf{a}}_{\mathbf{x}}(t) = \left( \frac{df(\mathbf{x}(t))}{d\mathbf{x}(t)} \right)^T \mathbf{a_x}(t)$$

However, to understand gradient-based learning, we require the parameter adjoint: $\mathbf{a}_{\boldsymbol{\theta}}(t) = \frac{dL(\mathbf{x}(T))}{d\boldsymbol{\theta}(t)}$. This can be accomplished by concatenating the original dynamical system by its parameters $\mathbf{z}(t) = [\mathbf{x}(t), \boldsymbol{\theta}(t)]$ (noting that $\dot{\boldsymbol{\theta}}(t) = 0$) to define the the augmented adjoint $\mathbf{a_z}(t) = [\mathbf{a_x}(t), \mathbf{a}_{\boldsymbol{\theta}}(t)]$, whose dynamics can be shown to follow

$$\dot{\mathbf{a}}_{\mathbf{z}}(t) = \left[ \left( \frac{df(\mathbf{x}(t))}{\mathbf{x}(t)} \right)^T \mathbf{a_x}(t), \left( \frac{df(\mathbf{x}(t))}{d\boldsymbol{\theta}} \right)^T \mathbf{a_x}(t) \right]$$

with terminal condition $\mathbf{a_z}(T) = \left[ \frac{dL(\mathbf{x}(T))}{d\mathbf{x}(T)}, 0 \right]$. The gradient of the loss with respect to the parameters can then be found by integrating the parameter adjoint dynamics backwards through time to get $\mathbf{a}_{\boldsymbol{\theta}}(0)$. In Supplementary Material E we provide a short derivation of the adjoint adapted from [38].

**Main results**. While the adjoint method is extensively used as a numerical tool in autodifferentiation packages [38, 40], it can also provide analytical insight into the gradient dynamics of dynamical systems. Towards this end, we now return to the case of an RNN, with the parameter of interest being the weight matrix. We demonstrate several mathematical results with the aid of the adjoint.

**Lemma 1.** *Consider the RNN $\dot{\mathbf{x}} = W\phi(\mathbf{x}) - \mathbf{x} + B\mathbf{u}(t)$, where $\mathbf{x}(t) \in \mathbb{R}^n$, $\mathbf{u}(t) \in \mathbb{R}^m$, $W \in \mathbb{R}^{n \times n}$, $B \in \mathbb{R}^{n \times m}$, and $\phi : \mathbb{R}^n \to \mathbb{R}^n$ an element-wise nonlinearity, and define a loss functional of the linearly decoded RNN state, $L(D\phi(\mathbf{x}(T)); \mathbf{y}(T))$ for $\mathbf{y} \in \mathbb{R}^d$. Furthermore, let $W = \sum_r^R \boldsymbol{\alpha}_r \otimes \boldsymbol{\beta}_r$. The adjoint dynamics are then given by,*

$$\dot{\mathbf{a}}_{\mathbf{x}} = \left( \sum_r^R \boldsymbol{\alpha}_r \otimes (\boldsymbol{\beta}_r \odot \phi'(\mathbf{x}(t))) \right)^T \mathbf{a_x} - \mathbf{a_x}, \qquad \dot{\mathbf{a}}_W = \mathbf{a_x} \otimes \phi(\mathbf{x}(t))$$

*with terminal conditions $\mathbf{a_x}(T) = \phi'(\mathbf{x}(T)) \odot \sum_{r'}^d L'(D_{r'} \cdot \phi(\mathbf{x}(T)); \mathbf{y}_{r'})D_{r'}$ and $\mathbf{a}_W(T) = 0$.* [2]

---

[2]Where $\odot$ denotes the Hadamard product and $\phi'$ the element-wise derivative of $\phi$

From the adjoint dynamics, it can be seen that the singular values of the gradient of the loss with respect to the weights of the RNN ($\nabla_W L = \mathbf{a}_W(0)$) depend on the dimensionality of the subspaces over which $\phi(\mathbf{x}(t))$ and $\mathbf{a_x}$ evolve. This can be used to show that the gradient's singular values are bounded by the singular values of both the activity and the adjoint, which we formalize in the following theorem.

**Theorem 1.** *Consider an RNN be defined as in Lemma 1. Then the singular values of its gradient can be bounded as:*

$$\max\left\{\sigma_n^{\phi(\mathbf{x})}\sigma_r^{\mathbf{a_x}}, \sigma_n^{\mathbf{a_x}}\sigma_r^{\phi(\mathbf{x})}\right\} \leq \sigma_r^{\nabla_W L} \leq \min\left\{\sigma_1^{\phi(\mathbf{x})}\sigma_r^{\mathbf{a_x}}, \sigma_1^{\mathbf{a_x}}\sigma_r^{\phi(\mathbf{x})}\right\},$$

*where $\sigma_r^{\mathbf{v}}$ denotes the $r$th singular value of $\mathbf{v}$.*

This theorem demonstrates that there is a natural bound to the numerical rank [41] of gradient-based learning in RNNs. Numerically, we find that the singular value spectrum of both $\phi(\mathbf{x})$ and $\mathbf{a_x}$ tend to decay exponentially, even in the case of a chaotic RNN (Fig. 6). In Supplementary Material E we repeat the same analysis with various activation functions, initial weight variances and ranks: overall, we find that smooth activation functions (such as the most commonly used tanh) tend to lead to the fastest decaying singular value spectrum of the adjoint.

So far, we haven't made any assumption on the architecture of the RNN such as constraints on $W$ or $\phi$. Under such constraints, stronger and more explicit bounds on both the matrix and tensor ranks can be obtained.

**Theorem 2.** *Consider an RNN defined as in Lemma 1 with $\phi =$id and $W^{(0)}$ of rank $R$. Furthermore suppose $x^{(k)}(0)$ is constrained to the subspace spanned by the columns of $W^{(k)}$ and $B$. Consider the weight tensor $\mathbf{W} = [W^{(0)}, W^{(1)}, ...]$ where $W^{(k+1)} = W^{(0)} + \alpha \sum_{j=1}^{k} \nabla_W L(D\mathbf{x}^{(j)}(T); \mathbf{y}^{(j)})$, with $\mathbf{y}^{(j)} \in \mathbb{R}^d$ where $\mathbf{x}^{(j)}$ denotes the activity of the RNN after the $j$th weight update. Then,*

1. *The rank of the gradient at the first step is at most $\operatorname{rank}(\nabla_W L^{(0)}) \leq R + 1$.*

2. *The rank of the trial slices of the weight tensor is at most $\operatorname{rank}(W^{(k)}) \leq 2R + m + d$.*

3. *The tensor rank of the weight tensor is at most $\operatorname{rank}(\mathbf{W}) \leq (2R + m + d)^2$.*

We note that these bounds are tight, in the sense that there exists networks for which they are equalities; therefore, they cannot be improved upon without restricting the set of architectures considered (e.g. to normal weight matrices). We also point out that they are non-trivial as the current best upper bound on the max rank of a $(\mathbb{R}^n)^{\otimes 3}$ tensor is $n^2 - n - 1$ [42]. In particular, in the case of $W^{(0)} = 0$ and $m = d = 1$ considered by [14], the tensor rank is at most 4. For arbitrary weight initializations, the matrix and tensor mathematical ranks can be high. Nevertheless, in most cases the matrix and tensor numerical ranks will fall vastly below these bounds due to Theorem 1.

We illustrate this result with $R = 3, d = 2, m = 2$ in Fig. 6. We find that the true ranks fall within these bounds – strictly below due to limited machine precision – and that the numerical ranks are extremely low. Intuitively, decoding the firing rate of the RNN into a low-dimensional space produces weight updates that push the RNN and adjoint activity to lie in a low-dimensional subspace of the state space. This, in turn, further pushes weight updates to lie in a low-dimensional subspace of the weight space.

The proofs of Lemma 1 and Theorems 1 and 2 are provided in Supplementary Material E. We also discuss two additional cases, namely that of a loss integrated over some period of time, and the gradient of the loss w.r.t. to other parameters of the RNN. Finally, we show that momentum-based optimization methods such as ADAM have the same property. Together, these analytical results provide insight as to why gradient dynamics bound the matrix rank of the weight updates in generic RNNs, as well as the tensor rank of learning dynamics in the linear case.

## 7  Discussion

**Summary**. In the present work, we explored the tensor rank of learning in artificial and biological neural networks. We showed that learning leads to low-tensor-rank weight updates, which can be exploited to uncover smooth changes in dynamics along with a principled choice of low-dimensional

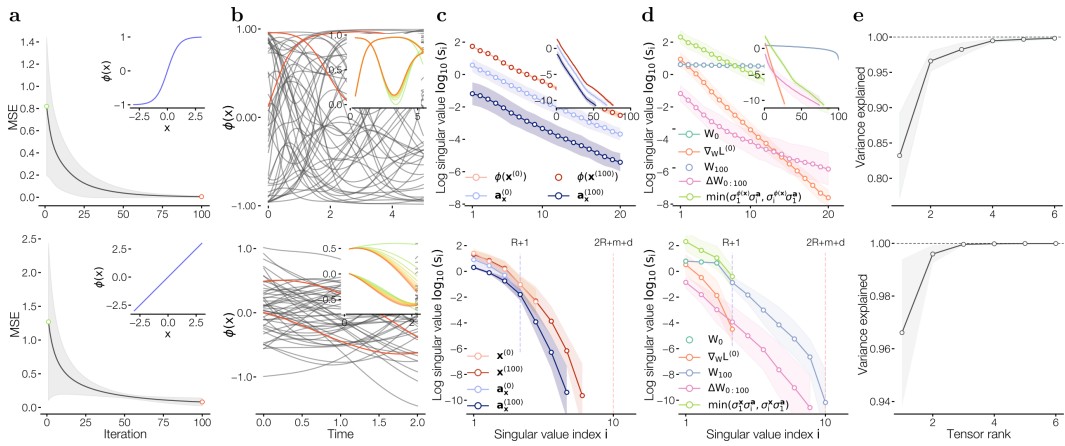

Figure 6: **The singular value spectrum of the gradient of random RNNs with random objective**. Top: Full rank RNN with tanh activation function in a chaotic regime. Bottom: Low-rank ($R = 3$) linear RNN. In both cases $m = d = 2$. The networks are trained to map a time-varying input (parameterized as an LDS) to a given target output. Error shade represents the standard deviation over 5 random initialization. **a**. Loss over training. **b**. Example of network activity at one iteration. Inset: activity of two neurons over learning, red is post-learning. **c**. Singular value spectrum of the activity and of the adjoint. **d**. Singular value spectrum of the weights, gradient, and bound we derive. **d**. Tensor rank of the weight tensor minus initial weights. Variance is computed per trial slice.

factors over which weights and activity evolve throughout the entire course of learning. Finally, we derived upper bounds on the singular values of gradient dynamics of nonlinear RNNs, and on the matrix and tensor ranks in the linear case.

**Modelling limitations and future work**. Inferring connectivity from neural recordings is in general an ill-posed problem [19]: given any observed pattern of activity, it is always possible that the data could be explained entirely in terms of a external input to a set of unconnected neurons. In our application to neural data, we resolved such ambiguities by assuming that inputs were fixed on each trial of a given condition, forcing the changes in neural activity across trials of the same condition to be captured by (smoothly varying) changes in weights. We note that, in principle, our framework and code could also implement residual trial-to-trial variability in the inputs.

In addition to learning- and condition-specific changes, neural recordings show substantial variability across consecutive trials, which are thought to reflect a combination of i) unmeasured covariates such as behavioural or intrinsic state, ii) stochasticity in the neural system itself, and iii) changes in the initial state at trial onset. Future work could incorporate behavioural covariates, allow for trial-specific initial states, and model the data using a stochastic dynamical system within each trial.

Here, we focused on gradient-based learning, motivated by recent evidence that it is able to explain many features of motor learning in neural data [17, 18], as well as by more general support for an optimization-based framework to understand neural learning [43]. The tensor rank of learning in RNNs with local synaptic rules (e.g., Hebbian) remains an open question. Towards this end, theoretical work has established links between Hebbian and gradient-based learning [44, 45], opening the possibility of an extension of our mathematical results to biologically plausible learning rules.

Our analytical results provide intuition for our observation that the rank of learning dynamics is limited by task complexity. This supports previous findings of low (matrix) rank weight changes in RNNs [14] and in deep networks [46, 47, 48]. Our results on the numerical matrix rank also has interesting ties to work that uses rank compression for more efficient training in deep networks [46], and for numerical solutions to systems with time-varying dynamics [49].

**Broader impact**. We introduce a novel framework for understanding learning in the brain and artificial neural networks. Our mathematical results on the rank of RNN gradients have broad relevance to the machine learning community, while our results on motor neural data could drive future applications to brain computer interfaces. Overall, our work makes novel contributions towards understanding the emergence of computations through learning in neural systems.

## Acknowledgments

We thank Francesca Mastrogiuseppe, Friedrich Schuessler, and Sina Tootoonian for feedback on the manuscript, and Matthias Hennig, Matt Nolan, and Srdjan Ostojic for helpful discussions. We are particularly grateful to Matthew Perich for sharing his data. This work was supported by the Agence Nationale de la Recherche (ANR-20-CE37-0004; ANR-17-EURE-0017).

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

# Supplementary Material

## A  Low tensor rank recurrent neural networks

### A.1  Architecture

**Low tensor rank weights**. In order to probe for the tensor rank of learning in neural data, we introduce an RNN architecture that captures the evolution of neural activity over slow timescales. We first recall the description of the architecture of the main text, $\mathbf{W} \in \mathbb{R}^{N \times N \times K}$,

$$\mathbf{W} = \sum_{r=1}^{R} \mathbf{a}_r \otimes \mathbf{b}_r \otimes \mathbf{c}_r.$$

In particular, at trial $k$, the dynamics of the RNN can be described as,

$$\tau \dot{\mathbf{x}} = W^{(k)} \phi(\mathbf{x}) - \mathbf{x} + B\mathbf{u}^{(k)}(t) = \sum_{r=1}^{R} c_r^{(k)} (\mathbf{a}_r \otimes \mathbf{b}_r) \phi(\mathbf{x}) - \mathbf{x} + B\mathbf{u}^{(k)}(t)$$

for $B \in \mathbb{R}^{N \times N_{\text{input}}}$, $\mathbf{u}^{(k)}(t) \in \mathbb{R}^{N_{\text{input}}}$, so that the RNN is a low rank RNN [15]. When training, we initialize $\mathbf{a}_r \sim \mathcal{N}(\mathbf{0}, I)$, $\mathbf{b}_r = \mathbf{a}_r$. Furthermore, the weights are parameterized such that $||\mathbf{a}_r|| = ||\mathbf{b}_r|| = 1$, so that the magnitude of a component is captured by $\mathbf{c}_r$.

**Smoothness constraint over trials**. We further constrain the initial covariance in trial of the trial factors by parameterizing them as $\mathbf{c}_r = (L + \sigma I)\bar{\mathbf{c}}_r$ where $LL^T = A$ is the Cholesky decomposition of the smooth covariance matrix $A$, and $\bar{\mathbf{c}}_r$ is initialized as $\bar{\mathbf{c}}_r \sim \mathcal{N}(\mathbf{0}, I)$. In particular, we use a rational quadratic kernel $s^2(1 + (2l)^{-1}(k_i - k_j)^2)^{-1}$, where $k_i$ is the $i$th trial index. This is equivalent to performing a 3-mode matrix-tensor product on the weight tensor itself $(L + \sigma I) \times_3 \mathbf{W}$ so that its entries over trials are linear combinations of smooth functions up to observation noise.

This parameterization is similar to that of Gaussian process regression, except no probabilistic objective is set (kernel regression). In particular, given that $(L + \sigma I)$ is invertible, any possible $\mathbf{c}_r$ can in theory be obtained upon optimization of the $\bar{\mathbf{c}}_r$. By additionally setting a regularization on $\bar{\mathbf{c}}_r$, we penalize the smoothness of $\mathbf{c}_r$ as non-smooth solutions have diverging $\bar{\mathbf{c}}_r$. In this way, we bias the optimization process towards smoother $\mathbf{c}_r$'s. As illustrated in Fig. 4b, the cross-validated loss remains similar to the non-smooth, full-rank, case.

A key advantage of having smooth trial factors is that missing trials can be easily accounted for. Indeed, in most large-scale neural datasets, such as the one explored in the present work, potentially many trials may have been discarded, for example due to behavioural performance being outside the range set by the experimentalist. The assumption being made here is that the across-trial covariance is preserved when such trials occur. That is, we assume that a failure of the animal to perform a given trial does not imply that a trial wasn't informative, or that learning did not occur.

**Condition-wise inputs**. We parameterize the condition-wise inputs to the network with neural ordinary differential equations [38], i.e. as a dynamical system whose right hand side is parameterized by a deep neural network (DNN)

$$\dot{\mathbf{v}}^{(i)} = DNN(\mathbf{v}^{(i)}) \quad \mathbf{u}^{(i)}(t) = \phi(D\mathbf{v}^{(i)}(t)),$$

where $\mathbf{v}^{(i)}(t) \in \mathbb{R}^{N_{\text{NODE}}}$ and $D \in \mathbb{R}^{N_{\text{input}} \times N_{\text{NODE}}}$. Throughout this work, we used a fully-connected 3-layer DNN with layers of size 150 and ReLU nonlinearities. This provides inputs whose dynamics are considerably less constrained than those generated by low-rank RNNs. Thus, we do not make any assumption on the activity of upstream brain regions which drive the activity of the brain region being recorded. While here we chose to model the inputs using autonomous neural ODEs, one might imagine feeding the neural ODE with behavioural or task covariates to relax the condition specificity assumption, or fitting residual inputs to capture trial-by-trial variability arising from unmeasured variables [21]. This could account for some of the variability that can neither be explained by the task condition, nor by changes in the dynamics due to learning.

**Loss**. In sections 4 and 5 we focus on optimizing for the mean squared error (MSE)

$$L(\mathbf{a}, \mathbf{b}, \mathbf{c}, B, M) = \sum_{k}^{K} \sum_{q}^{T} \left|\left| M\phi(\mathbf{x}^{(k)}(t_q)) - \mathbf{y}^{(k)}(t_q) \right|\right|^2, \tag{A.1.1}$$

where $\mathbf{y}$ are firing rate estimates data and $M \in \mathbb{R}^{N \times N_{\text{data}}}$. In section B we present supplementary results on optimizing the Poisson log-likelihood with respect to spike data,

$$L(\mathbf{a}, \mathbf{b}, \mathbf{c}, B, M) = -\sum_{k}^{K} \sum_{q}^{T} \log(\text{Poisson}(\mathbf{y}^{(k)}(t_q) | M\phi(\mathbf{x}^{(k)}(t_q)))) \tag{A.1.2}$$

where $\mathbf{y}$ are binned spike data.

**Pseudocode.** The following pseudocode summarizes the steps of fitting an ltrRNN to neural data. For the sake of clarity, we present the simplest case: an autonomous ltrRNN with fixed initial state. The neural ODE-driven case can be achieved by coupling $f$ with a deep neural net. Additionally, to allow for the *initial state* (w.r.t. the data) of the RNN to change over conditions and trials, the evaluation of the dynamical system can be done over $\{-T_0\Delta t, ..., 0, ..., T\Delta t\}$, where $\mathbf{x}_0$ is now the state at $-T_0\Delta t$, and the fit to data is still done on the non-negative time states.

---

**Algorithm 1** Low tensor rank recurrent neural network fit to data

**inputs:**
    $\mathbf{Y} \in \mathbb{R}^{N_{\text{data}} \times T \times K}$                $\triangleright$ Neural data tensor of shape neuron $\times$ time $\times$ trial
    $\mathbf{t} = \{0, \Delta t, ..., T\Delta t\}$              $\triangleright$ Time points of evaluation of the dynamical system

**initializations:**
    randomly initialize $\mathbf{a}_r, \mathbf{b}_r \in \mathbb{R}^N$ for $r = 1, \dots, R$
    randomly initialize $\bar{\mathbf{c}}_r \in \mathbb{R}^N$ for $r = 1, \dots, R$
    randomly initialize $\mathbf{q} \in \mathbb{R}^R$, $M \in \mathbb{R}^{N_{\text{data}} \times N}$

**definitions:**
    $f_W(\mathbf{x}) = W\phi(\mathbf{x}) - \mathbf{x}$              $\triangleright$ RNN dynamics for a given weight matrix
    $A \in \mathbb{R}^{K \times K}$ such that $A_{ij} \leftarrow \kappa(i, j)$     $\triangleright$ Trial covariance matrix defined by a smooth kernel $\kappa$
    $L \leftarrow \text{Cholesky}(A + \sigma^2 I)$                 $\triangleright$ Cholesky decomposition

**while** the loss $l$ hasn't converged **do**
    $l \leftarrow 0$
    $\mathbf{c}_r \leftarrow L\bar{\mathbf{c}}$
    $\mathbf{W} \leftarrow \sum_{r=1}^{R} \mathbf{a}_r \otimes \mathbf{b}_r \otimes \mathbf{c}_r$
    $\mathbf{x}_0 \leftarrow \sum_{r=1}^{R} q_r \mathbf{a}_r$                       $\triangleright$ The initial state
    **for** $k = 1, ..., K$ **do**                 $\triangleright$ For all trials (parallelizable)
        $X^{(k)} \leftarrow \text{ODESolve}(f_{W^{(k)}}, \mathbf{x}_0, \mathbf{t})$   $\triangleright$ Matrix of activity of the RNN during trial $k$
        $l \leftarrow l + \underbrace{||M\phi(X^{(k)}) - Y^{(k)}||_2^2}_{\text{Fit to data}} + \underbrace{\alpha||X^{(k)}||_2^2}_{\text{Regularization}}$
    **end for**
    $\text{SGDUpdate}(\mathbf{a}_r, \mathbf{b}_r, \bar{\mathbf{c}}_r, M, \mathbf{q})$              $\triangleright$ E.g. $\mathbf{a}_r \leftarrow \mathbf{a}_r - \eta \frac{dl}{d\mathbf{X}} \frac{d\mathbf{X}}{d\mathbf{W}} \frac{d\mathbf{W}}{d\mathbf{a}_r}$
**end while**

---

**Code availability**. The ltrRNN implementation can be found at https://github.com/arthur-pe/LtrRNN.

## A.2 The dynamics of ltrRNNs

**Rich changes of dynamics through oblique columns**. Unlike for a matrix rank decomposition, a tensor rank decomposition can be unique even for non-orthogonal factors. A sufficient condition for uniqueness is that $r_\mathbf{a} + r_\mathbf{b} + r_\mathbf{c} \leq R - 2$ where, without loss of generality, $r_\mathbf{a}$ denotes the maximum number of linearly dependent columns of $\mathbf{a}$ [50]. In other words, fitting the changes of dynamics over trials as opposed to a single low rank RNN shared over all trials gives additional information regarding the columns and rows. In the case where the $\mathbf{a}_j$'s are not orthogonal, non-trivial qualitative changes in the vector fields can occur. Since for any given trial $k$, an ltrRNN is simply a low rank RNN [15],

$$\dot{\mathbf{x}}^{(k)} = \sum_{j} \mathbf{a}_j (c_j^{(k)} \mathbf{b}_j \cdot \phi(\mathbf{x}^{(k)})) - \mathbf{x}^{(k)} + B\mathbf{u}^{(k)}(t) \tag{A.2.1}$$

Table 1: **Hyperparameters of the ltrRNN models**. Bold indicates values specific to section 4 and 5, * indicates cross-validated hyperparameters, other hyperparameters were tuned by hand.

|  |  | Neural data (S4) | Simulated data (S5) |
|---|---|---|---|
| **LtrRNN** | | | |
| | $R$ | 5* | 5* |
| | $n$ | 200* | 200* |
| | $\phi$ | tanh | tanh |
| **Smoothness** | | | |
| | $l$ | **50** | **15** |
| | $s$ | 0.1 | 0.1 |
| | $\sigma$ | 0.1 | 0.1 |
| **Neural ODE** | | | |
| | Layers | **$150 \times 150 \times 150$** | **N/A** |
| | $\phi$ | **ReLU** | **N/A** |
| **Regularization** | | | |
| | $\alpha$ | **0.01** | **0** |
| **Cross-validation** | | | |
| | Train blocks | $1 \times 10 \times \mathbf{20}$ | $1 \times 10 \times \mathbf{10}$ |
| | Test blocks | $1 \times 5 \times \mathbf{10}$ | $1 \times 5 \times \mathbf{5}$ |
| | | | neuron $\times$ time $\times$ trial |

Table 2: **Approximate training time of ltrRNNs**. Here on the neural data of section 4. The variables which impact the most training time are the trial and neuron dimensions of the ltrRNN (not the rank or data time steps), as well as the neural ODE architecture. Hardware : desktop with an RTX 3090 Nvidia GPU and i7-12700K Intel CPU. * indicates the one used in section 4.

|  |  | Neurons | |
|---|---|---|---|
| | | **200** | **400** |
| **Trials** | **370** | 12min | 22min |
| | **740** | 16min* | 40min |

so that the dynamics of $\mathbf{x}^{(k)}$ are constrained to $\mathrm{span}\{\mathbf{a}_j\} \cup \{B_j\}$. Unlike low rank RNNs, ltrRNN are not necessarily invariant under changes of bases of $\mathbf{a}_j$'s. Nevertheless, we can introduce an orthonormal basis $\{\tilde{\mathbf{a}}_j\}$ so that,

$$W^{(k)}\phi(\mathbf{x}^{(k)}) = \sum_i^R \tilde{\mathbf{a}}_i \sum_j (\tilde{\mathbf{a}}_i \cdot \mathbf{a}_j)(\mathbf{b}_j \cdot \phi(\mathbf{x}))c_j^{(k)} \qquad (A.2.2)$$

In particular, notice that the dynamics along all $\tilde{\mathbf{a}}_i$ could potentially be affected by varying $c_j^{(k)}$.

Conversely, the $\mathbf{a}_i$'s and $\mathbf{b}_i$'s being respectively orthonormal — e.g. as in a singular value decomposition — is not a sufficient condition for uniqueness of the tensor rank decomposition [51]. Nevertheless, constraining the $\mathbf{a}_i$ to be orthogonal and ensuring that the Kruskal constraint is satisfied, the vector fields are then orthogonal. In that case, varying $c_i^{(k)}$ for some $i$ corresponds to rescaling the vector field along $\mathbf{a}_i$. Nevertheless, as is illustrated below with a single component, the leak term allows the system to display typical properties of linear and nonlinear dynamical systems.

**Bifurcation in a tensor rank-one RNN**. A classical example of bifurcation in a two-neuron system is that of the pitchfork supercritical bifurcation. Here, we show that it is essentially a tensor rank one RNN. We also illustrate how the corresponding linear RNN bifurcates. Let $\mathbf{a} = \mathbf{b} = [-1/\sqrt{2}, 1/\sqrt{2}]^T$ so that,

$$W^{(k)} = \frac{c}{2}\begin{bmatrix} 1 & -1 \\ -1 & 1 \end{bmatrix} \xrightarrow{\dot{\mathbf{x}}=0} \begin{bmatrix} \mathbf{x}_1 \\ \mathbf{x}_2 \end{bmatrix} = \frac{c}{2}\begin{bmatrix} \phi(\mathbf{x}_1) - \phi(\mathbf{x}_2) \\ -\phi(\mathbf{x}_1) + \phi(\mathbf{x}_2) \end{bmatrix} \qquad (A.2.3)$$

That is $\mathbf{x}_1 = -\mathbf{x}_2$. We consider two cases, $\phi = \tanh$ and $\phi = \mathrm{id}$, both odd functions. Introducing the two in the previous equation, $-\phi(\mathbf{x}_2) = \phi(\mathbf{x}_1)$. So that $\mathbf{x}_i = c\phi(\mathbf{x}_i)$. Now considering each activation function separately,

- tanh: i) $c > 1$ has two solutions. ii) $c \leq 1$ one solution (the origin). At $c = 1$ the origin is a non-hyperbolic fixed point.
- id: i) $c = 1$ for any $\mathbf{x}_1$. The non-zero eigenvalue of the Jacobian of the system is negative, therefore it is a line attractor. ii) $\mathbf{x}_1 = 0$ for any $c$. Then the Jacobian of the system at the origin has both positive and negative eigenvalues for $c > 1$ and only negative for $c \leq 1$.

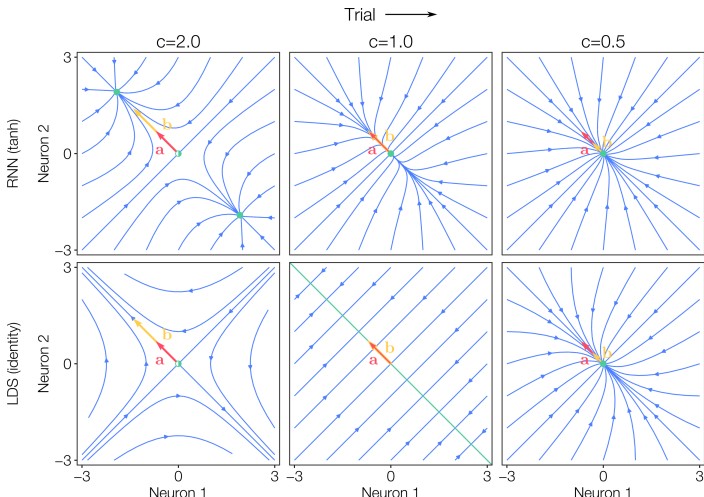

Supplementary Figure 1: **Bifurcation in a tensor rank one RNN**.

**More general changes in vector fields**. At the cost of increasing the rank, an ltrRNN can possibly transition between two arbitrary vector fields of ranks $R_1$ and $R_2$. For example, let $\mathbf{c}_j = [1, 0.9, ..., 0]$ for $j \in \{1, ..., R_1\}$ and $\mathbf{c}_j = [0, ..., 0.9, 1]$ for $j \in \{R_1 + 1, ..., R_1 + R_2\}$. There might however be multiple bifurcations between the first and last trial. More generally, given that any tensor has a (possibly high rank) tensor decomposition, any weight tensor can in theory be captured by an ltrRNN. This further illustrates the relevance of the result we found that ltrRNN of very low ranks fit data as well as full tensor RNNs.

### A.3 Relationship between rank decomposition and eigendecomposition

In this section, we investigate the relationship between an arbitrary low rank decomposition of a matrix and the eigendecomposition. We assume that, on a given trial $k$, $W^{(k)}$ has the rank decomposition

$$W^{(k)} = \mathbf{a}\,\mathrm{diag}(\mathbf{c}^{(k)})\mathbf{b}^T = \sum_{i=1}^{R'} c_i^{(k)} \mathbf{a}_i \mathbf{b}_i^T. \tag{A.3.1}$$

where $\mathbf{a} \in \mathbb{C}^{N \times R'}$, $\mathbf{b} \in \mathbb{C}^{R' \times N}$, $\mathbf{c}^{(k)} \in \mathbb{R}^{R'}$, and $\mathbf{a}_i, \mathbf{b}_i$ are the rows/colums of $\mathbf{a}, \mathbf{b}$ respectively. One such low rank decomposition is the eigendecomposition[3][4]

$$W^{(k)} = V\Lambda V^{-1} = V\,\mathrm{diag}(\boldsymbol{\lambda})V^{-1} = \sum_{i=1}^{R} \lambda_i \mathbf{v}_i \tilde{\mathbf{v}}_i^T.$$

By the rank-nullity theorem, $R$ is the rank of $W^{(k)}$, so that $R' \geq R$, with $R = R'$ when Equation (A.3.1) is a minimal rank decomposition. Equating the two decompositions gives a general expression for the eigenvalues:

$$\Lambda = V^{-1}\mathbf{a}\,\mathrm{diag}(\mathbf{c})\mathbf{b}^T V \implies \lambda_i = (V^{-1}\mathbf{a}\,\mathrm{diag}(\mathbf{c})\mathbf{b}^T V)_{ii} = \sum_j c_j (\tilde{\mathbf{v}}_i \cdot \mathbf{a}_j)(\mathbf{v}_i \cdot \mathbf{b}_j).$$

---

[3]Assuming $W^{(k)}$ is diagonalisable. A similar argument using the Jordan normal form holds for defective matrices.

[4]Here, we write the left eigenvectors (rows of $V^{-1}$, transposed into column vectors) as $\tilde{\mathbf{v}}_r$. We can assume without loss of generality that the right eigenvectors $\mathbf{v}_i$ are normalised to unit length, in which case the orthonormality of left and right eigenvectors gives $\tilde{\mathbf{v}}_i \cdot \mathbf{v}_j = \delta_{ij} = \|\tilde{\mathbf{v}}_i\|\|\mathbf{v}_j\|\cos(\theta_{ij}^{\tilde{\mathbf{v}},\mathbf{v}}) \implies \|\tilde{\mathbf{v}}_i\| = 1/\cos(\theta_{ii}^{\tilde{\mathbf{v}},\mathbf{v}})$, where $\theta_{ii}^{\tilde{\mathbf{v}},\mathbf{v}}$ is the angle between the $i$th left and right eigenvector.

If $\lambda_i, \mathbf{a}, \mathbf{b}, \mathbf{c}^{(k)} \in \mathbb{R}$, this gives rise to the result stated in the main text:

$$\lambda_i = \sum_j c_j (\|\tilde{\mathbf{v}}_i\| \|\mathbf{a}_j\| \cos \theta_{ij}^{\tilde{\mathbf{v}}, \mathbf{a}})(\|\mathbf{v}_i\| \|\mathbf{b}_j\| \cos \theta_{ij}^{\mathbf{v}, \mathbf{b}}) \tag{A.3.2}$$

$$= \sum_j c_j \left( \frac{\|\mathbf{a}_j\|}{\cos(\theta_{ii}^{\tilde{\mathbf{v}}, \mathbf{v}})} \cos \theta_{ij}^{\tilde{\mathbf{v}}, \mathbf{a}} \right)(\|\mathbf{b}_j\| \cos \theta_{ij}^{\mathbf{v}, \mathbf{b}}) \tag{A.3.3}$$

$$= \sum_j c_j \frac{\|\mathbf{a}_j\| \|\mathbf{b}_j\|}{\cos(\theta_{ii}^{\tilde{\mathbf{v}}, \mathbf{v}})} \cos \theta_{ij}^{\tilde{\mathbf{v}}, \mathbf{a}} \cos \theta_{ij}^{\mathbf{v}, \mathbf{b}} \tag{A.3.4}$$

$$= \frac{1}{2} \sum_j c_j \frac{\|\mathbf{a}_j\| \|\mathbf{b}_j\|}{\cos(\theta_{ii}^{\tilde{\mathbf{v}}, \mathbf{v}})} (\cos(\theta_{ij}^{\tilde{\mathbf{v}}, \mathbf{a}} + \theta_{ij}^{\mathbf{v}, \mathbf{b}}) + \cos(\theta_{ij}^{\tilde{\mathbf{v}}, \mathbf{a}} - \theta_{ij}^{\mathbf{v}, \mathbf{b}})). \tag{A.3.5}$$

Note that the above derivation makes no assumptions about the form of the low rank decomposition, other than that it is real-valued. Low rank decompositions commonly set $\|\mathbf{a}_i\| = \|\mathbf{b}_i\| = 1$, and often enforce orthogonality on the $\mathbf{a}_i$ and/or $\mathbf{b}_i$, thereby introducing additional constraints on the relationship between the $c$'s and $\lambda$'s. Normal matrices have $\cos(\theta_{ij}^{\tilde{\mathbf{v}}, \mathbf{v}}) = \delta_{ij}$ and $\tilde{\mathbf{v}}_i = \mathbf{v}_i$, in which case further simplifications can be made.

## B  Motor learning

**Pre-processing**. Motor ($n = 72$ for Fig. 4, $n = 70$ for Sup. Fig. 4) and premotor ($n = 231$ for Fig. 4, $n = 137$ for Sup. Fig. 4) cortical neurons were used. The data were Gaussian filtered with a standard deviation of 40 ms (4 time bins). It was then centered by its baseline activity through subtracting neuron-wise the mean activity from around target onset to go-cue, and rescaled by dividing neuron-wise by the standard deviation of the execution period. Namely, the activity of a neuron $\bar{\mathbf{y}}_i(t)$ was given by

$$\bar{\mathbf{y}}_i(t) = \frac{\mathbf{y}_i(t) - \langle \mathbf{y}_i(t) \rangle_{t \leq 100}}{\langle (\mathbf{y}_i(t) - \langle \mathbf{y}_i(t) \rangle_{t > 100})^2 \rangle_{t > 100}} \tag{B.0.1}$$

where $t = 0$ is the go cue. Example of activity upon this pre-processing is given in Supp. Fig. 2.

**Modeling assumptions**. We assumed that motor and premotor cortex was driven into an initial state by inputs from upstream regions during the preparatory period, after which the input shuts off so that the resulting activity during the reach evolves autonomously via the recurrent dynamics dynamics from that initial state. We therefore set $\mathbf{u}(t) = 0$ for $t > 100$ where $t = 0$ is the time of the go cue. Where the 100 ms account for a sensory delay.

**Cross-validation procedure**. We cross-validated the optimal rank and number of neurons of the ltrRNN. Low matrix or tensor rank models can be cross-validated by holding out specific entries of the matrix or tensor for training, and then used for testing. However, neural data has temporal correlation, such that the entry of the time-by-trial-by-neuron data tensor $T_{ijk}$ is strongly correlated with $T_{i-1,jk}$ and $T_{i+1,jk}$. For example, assuming the data are continuous, a simple average of these entries will give an optimal estimate $T_{ijk}$ in the limit of small time bins. Thus, the test set can be trivially inferred from the train set. We validated this intuition by performing the same cross-validation as section 4 but with $1 \times 1 \times 1$ blocks, and found the test loss was similar as using the train loss over the whole dataset (Sup. Fig. 3).

To counter this effect, sets of contiguous entries $[T_{ijk}, ..., T_{i+n,jk}]$ can be held out of training, and the interior of these blocks $[T_{i+q,jk}, ..., T_{i+n-q,jk}]$ used for testing [30]. Here, given that we are interested in uncovering smooth changes in neural activity over slow timescales, we hold out $n$-by-$m$ matrices $[[T_{ijk}, ..., T_{i+n,jk}], ..., [T_{i,j+m,k}, ..., T_{i+n,j+m,k}]]$ (Fig. 4b. inset).

As mentioned in Supplementary Material A, our method allows inferring the dynamics of held-out trials. We found that using the cross-validation procedure from [30] infers similar ranks as holding out entire trials (Sup. Fig. 3). We nevertheless applied this procedure for the sake of being able to compare different classes of models.

**Poisson log likelihood**. Another method for fitting firing-rate models to spike data is to use the negative Poisson log likelihood loss [11, 10]. We fitted a ltrRNN ($R = 5$, $n = 200$ as in the MSE

loss case) with softplus activation using negative Poisson log likelihood loss. We found similar but overall noisier results (Supp. Fig. 5). This may be due to the presence of high firing rate neurons which are normalized by the preprocessing procedure but not likelihood fitting.

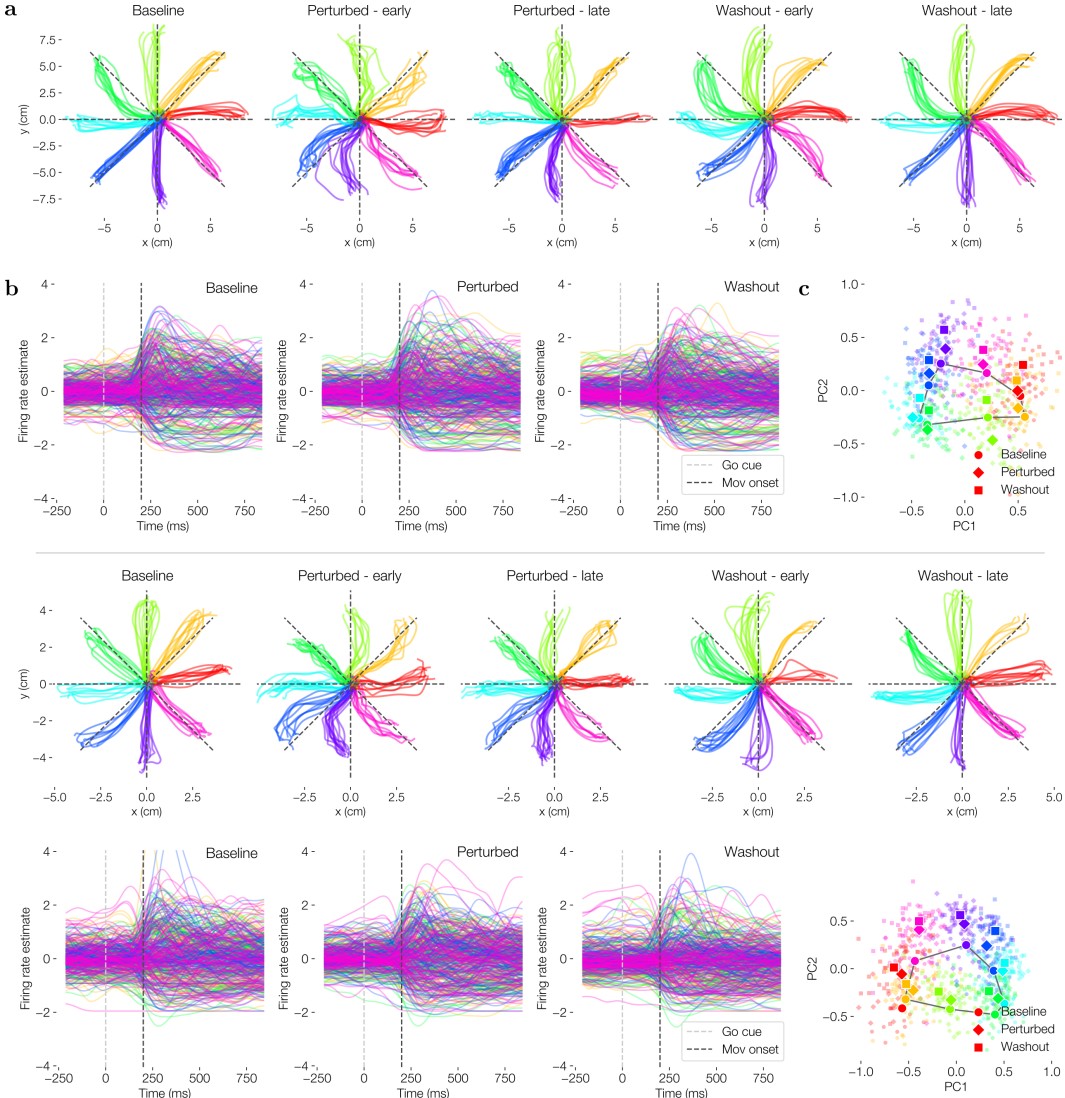

Supplementary Figure 2: **Persistent effects of motor learning**. Top: session used in the main text (Fig. 4). Bottom : session used in supplementary material (Sup. Fig. 4). **a**. Hand movement during the first and last 80 trials of perturbation learning and washout. The hand trajectories of some reach directions do not revert back post-washout (e.g. light green for top; dark green for bottom) **b**. Single neuron activity averaged within each condition. **c**. State at go cue $+100$ms. Larger full color marker are median within a condition. For some reach directions, the washout tends to be more similar to the perturbed state than the baseline.

## C    Task-trained RNN model of motor learning

**Task design**. A trial is split into a preparatory period $t \in [0, T_{go})$ and execution period $[T_{go}, T_{end}]$. Here we set $T_{go} = 2$, $T_{end} = 4$. To model a ballistic reach, the RNN receives input the target information and a hold cue during the preparatory period. During the execution period it evolves

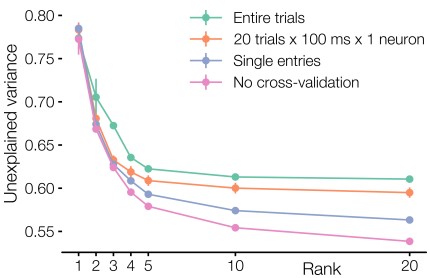

Supplementary Figure 3: **Comparison of cross-validation procedures**. Applied to the neural data of the session used in the main text.

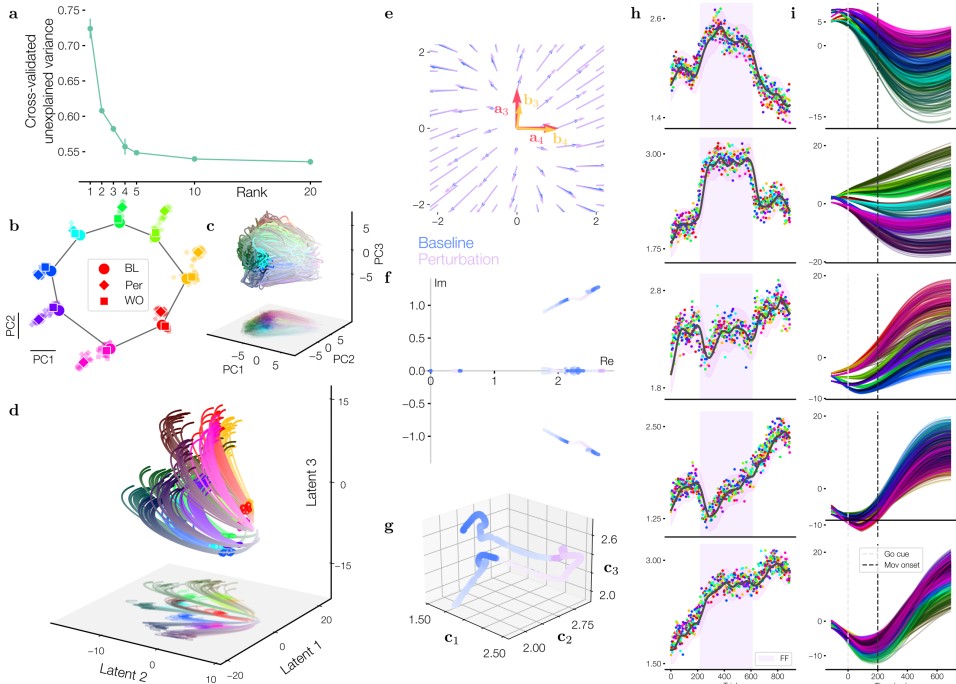

Supplementary Figure 4: **LtrRNN applied to an additional recording session**. **a**. Cross-validated loss with held out blocks of size 100ms by 20 trials. **b**. PCA on preprocessed data. **c**. State of the ltrRNN at go cue +100ms. **f**. Projection on first three $\mathbf{a}_j$. **e**. Projection of the vector field along $\mathbf{a}_j$ (see main text). **f**. Eigenspectrum of $W^{(k)}$ over trials. **g**. First three $\mathbf{c}_j$. **h**. Trial factors $\mathbf{c}_j$. **i**. Projection of $\mathbf{x}^{(k)}(t)$ on the corresponding $\mathbf{a}_j$.

autonomously.

$$\mathbf{dx}^{(k)} = \left[ W\phi(\mathbf{x}^{(k)}) - \mathbf{x}^{(k)} + \mathbb{1}_{t<T_{go}} B_{target} \mathbf{u}_{target}^{(k)} + \mathbb{1}_{t<T_{go}} B_{hold} \mathbf{u}_{hold}^{(k)} \right] dt + \sigma \mathbf{dW} \quad \text{(C.0.1)}$$

where $B_{target} \in \mathbb{R}^{n \times 2}$, $\mathbf{u}_{target}^{(k)} = [\cos(\theta^{(k)}), \sin(\theta^{(k)})]$ is a static vector representing the position of the target, $B_{hold} \in \mathbb{R}^{n \times 1}$, $\mathbf{u}_{hold}^{(k)} = 1$ a cue indicating to hold movement, and $\mathbf{dW}$ the infinitesimal increments of a Wiener process [52]. The dynamics of the hand are given in section 5 of the main text. The loss is taken to minimize the distance between the hand $\mathbf{y}^{(k)}(t)$ and the target $\mathbf{v}^{(k)}$ throughout the execution period, while keeping the hand still during the preparatory period, that is $L(W, B_{target}, B_{hold}, D) =$

$$\frac{1}{K} \sum_{k=1}^{K} \left( \frac{1}{T_{go}} \int_0^{T_{go}} ||\mathbf{y}^{(k)}(t)||^2 dt + \frac{1}{T_{end} - T_{go}} \int_{T_{go}}^{T_{end}} ||\mathbf{y}^{(k)}(t) - \mathbf{v}^{(k)}||^2 dt \right). \quad \text{(C.0.2)}$$

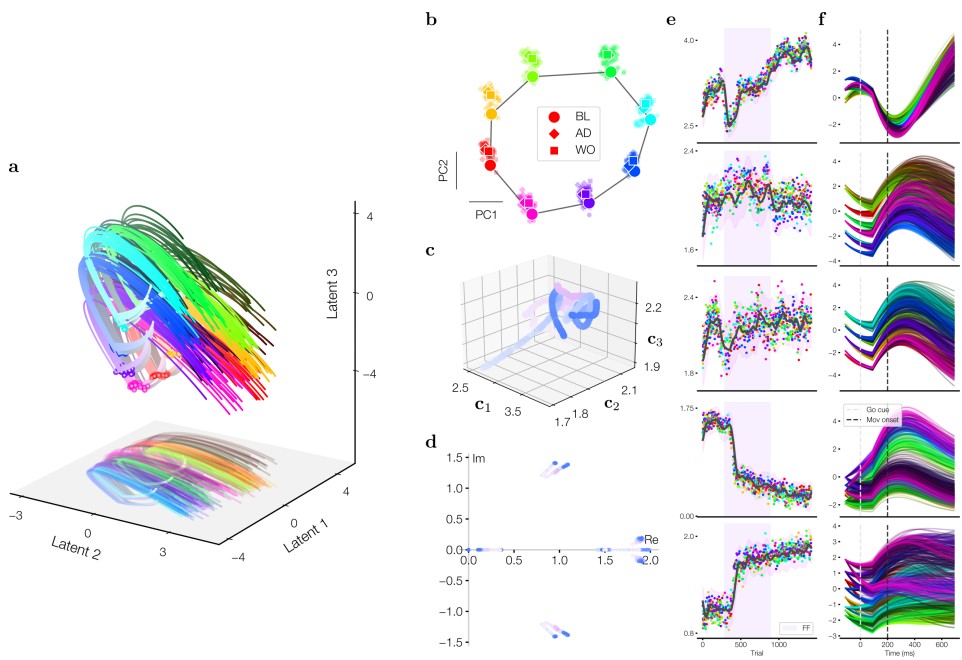

Supplementary Figure 5: **Poisson log-likelihood fitting**. **a**. Projection on first three $\mathbf{a}_j$. **b**. State of the ltrRNN at go cue +100ms. **c**. First three $\mathbf{c}_j$. **d**. Eigenspectrum of $W^{(k)}$ over trials. **e**. Trial factors $\mathbf{c}_j$. **f**. Projection of $\mathbf{x}^{(k)}(t)$ on the corresponding $\mathbf{a}_j$.

In particular, the speed of the reach is only constrained by the noise of the RNN and the hand. The dynamical system as a whole is evaluated with a differentiable adaptive step SDE solver [40] and trained with ADAM [53] during initial training, and SGD during motor perturbation learning.

**Analysis of the weights**. We found that, consistent with the literature [17], the changes in weights resulting from the initial training were much larger than those of motor perturbation learning. PARAFAC on the full tensor of updates $\mathbf{W} - W_0 \otimes \mathbf{1}$ captured the weight tensor in 3 components (not shown), whose columns and rows were essentially those of performing SVD on $W^*$. Fitting additional PARAFAC components revealed that residual variability in the updates of pretraining was larger than the motor perturbation learning variability. Furthermore, unlike SVD, there is no guarantee that the components of fitting a rank $k + 1$ PARAFAC model will be related to those of fitting a rank $k$ model. Nevertheless, motor perturbation learning had a significant change on the eigenvalues and activity of the RNN (Fig. 5f,g)

To uncover an upper bound on rank of the weight tensor, we split the analysis into the pre-training and motor perturbation learning. We first performed SVD on the change in weights matrix $W^* - W_0$, where $W^*$ are the weights of the network post-training, but pre-motor perturbation learning (Fig. 5b.). We found that the changes in weights were well captured by a rank-3 decomposition. Then, we performed PARAFAC on the change in weights tensor $\mathbf{W} - W^* \otimes \mathbf{1}$ of the motor perturbation learning. We found that this change of weight tensor was well approximated by a tensor rank 2 decomposition. The combination of these results upper-bounds the tensor rank of the overall changes in weights to 5. Finally, we compared the subspace spanned by the columns of the SVD and PARAFAC decomposition by projecting the weight tensor $\mathbf{W} - W^* \otimes \mathbf{1}$ on the first three column and row singular vectors of $W^* - W_0$ and found approximately a remaining 0.2 unexplained variance, suggesting that the columns of $\mathbf{W}_0 - W^*$ and $W^*$ were not orthogonal, but did not span the same subspace. Combined, these results suggest that the numerical tensor rank is at least 4 and at most 5, consistent with the results uncovered by ltrRNN from the RNN activity (Fig. 5e).

# D   Task-trained RNN models of additional neuroscience tasks

To investigate the generality of the low tensor rank framework, we additionally trained RNNs on three non-motor tasks commonly used in neuroscience.

**Sensory evidence accumulation task** [54] (Fig. 6i). The RNN receives a one-dimensional Ornstein-Uhlenbeck (OU) process input whose expected steady-state is either positive or negative. The target output is either +1 or -1 if the mean of the input is positive or negative, respectively.

**Contextual decision making task** [10] (Fig. 6ii). The RNN receives a three-dimensional input. The first two inputs are independent OU processes as in the previous task. The third input is binary and constant, and determines which of the two stochastic inputs must be integrated. The target output is +1 (or -1) if the mean of the input of the OU process indicated by the contextual input is positive (or negative).

**Working memory task** [14] (Fig. 6iii). The RNN receives a 1-dimensional input consisting of two stimuli of different amplitudes separated by a delay in time. The target output is the identity of the stimulus (1 or 2) which had larger amplitude.

Note that, in contrast to the motor adaptation task, these tasks do not contain any baseline period. Therefore, as we could not compare weight updates to a pre-trained solution, we simply analyzed the tensor rank of the weights starting from random initialization. That is, we first trained an RNN on each of these tasks using gradient descent, then used PARAFAC on $\mathbf{W} - W^{(0)} \otimes \mathbf{1}$ to determine the tensor rank of the resulting neuron $\times$ neuron $\times$ iteration tensor of weights over training. In each of these tasks we found that the variance explained indeed saturated at low tensor ranks (at $R = 1, 4$, and 3; Fig. 6e).

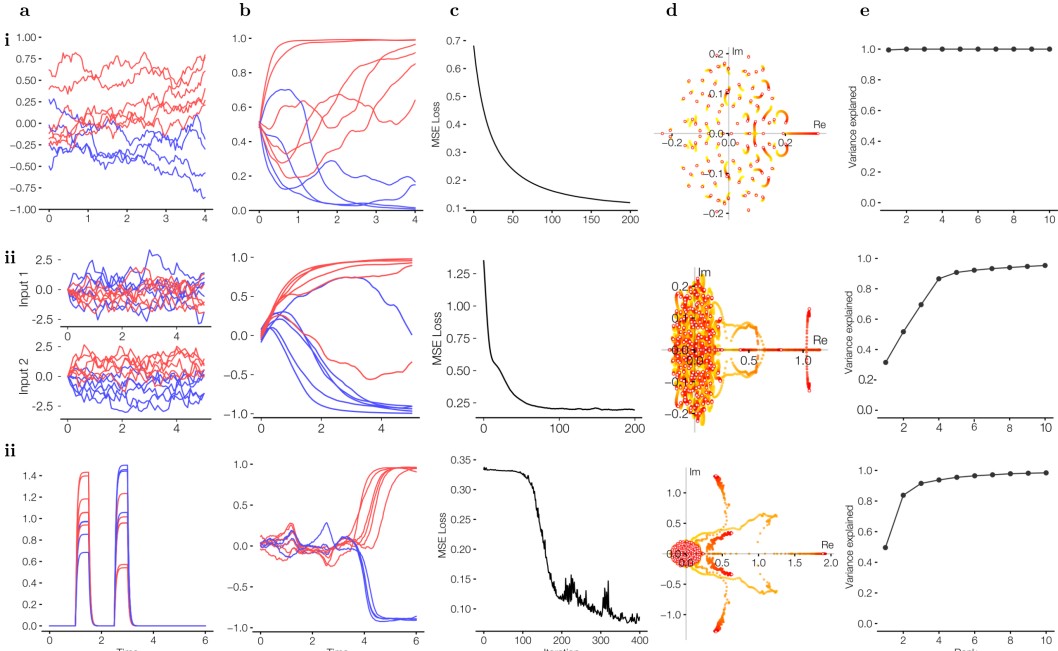

Supplementary Figure 6: **Validation of low-tensor-rank framework on RNNs trained on additional tasks.** Each row indicates a different task. **Row i.** Sensory evidence accumulation. **a.** Example inputs. Each line represents a different trial; red/blue indicate target identity. **b.** Example outputs after training (for the inputs shown in **a**). **c.** Loss curve over iterations. **d.** Eigenspectrum of $W^{(k)} - W^{(0)}$. Color gradient indicates training iterations $k$ (with $k = 1$ in yellow). **e.** After training, we perform PARAFAC on $\mathbf{W} - W^{(0)} \otimes \mathbf{1}$ to estimate the rank of learning dynamics. **Row ii.** Contextual decision making. **Row iii.** Working memory task.

# E  Mathematical results

## E.1  Adjoint derivation

In this section, we present a derivation of the adjoint mainly following [38]. Then, we derive the adjoint of recurrent neural networks.

### E.1.1  State adjoint

Consider the dynamical system $\dot{\mathbf{x}} = f(\mathbf{x}, \boldsymbol{\theta}) \in \mathbb{R}^n$ where $\boldsymbol{\theta} \in \mathbb{R}^k$ is a set of parameters. Furthermore, let the functional $L : \mathbb{R}^n \to \mathbb{R}$ such that $L(\mathbf{x}(T))$ is our loss. First, define the *state adjoint*,

$$\mathbf{a}(t) = \frac{dL(\mathbf{x}(T))}{d\mathbf{x}(t)}. \tag{E.1.1}$$

Notice that since $\mathbf{x}(t + \Delta t)$ is a function of $\mathbf{x}(t)$,

$$\mathbf{a}(t) = \frac{dL(\mathbf{x}(T))}{d\mathbf{x}(t + \Delta t)} \frac{d\mathbf{x}(t + \Delta t)}{d\mathbf{x}(t)} = \mathbf{a}(t + \Delta t) \frac{d\mathbf{x}(t + \Delta t)}{d\mathbf{x}(t)}. \tag{E.1.2}$$

By Taylor expanding $\mathbf{x}(t + \Delta t) = \mathbf{x}(t) + \Delta t f(\mathbf{x}(t)) + O(\Delta t^2)$, we get,

$$\mathbf{a}(t) = \mathbf{a}(t + \Delta t)(I + \frac{d}{d\mathbf{x}(t)} f(\mathbf{x}(t)) + O(\Delta t^2)) \tag{E.1.3}$$

where $I$ is the identity matrix. By rearranging,

$$\frac{\mathbf{a}(t + \Delta t) - \mathbf{a}(t)}{\Delta t} = \mathbf{a}(t + \Delta t) \frac{df(\mathbf{x}(t))}{d\mathbf{x}(t)} + O(\Delta t). \tag{E.1.4}$$

Taking the limit as $\Delta t \to 0$,

$$\frac{d\mathbf{a}(t)}{dt} = \mathbf{a}(t) \frac{df(\mathbf{x}(t))}{d\mathbf{x}(t)}. \tag{E.1.5}$$

We now have the dynamics of the adjoint; all that remains is that we find an initial (or rather terminal) condition. For this, notice that

$$\mathbf{a}(T) = \frac{dL(\mathbf{x}(T))}{d\mathbf{x}(T)} \tag{E.1.6}$$

is the usual gradient of $L$ w.r.t. to its argument.

### E.1.2  Parameter adjoint and gradient

In the above we derived the state adjoint $\mathbf{a}$. However, for our purposes we also require the *parameter adjoint* $dL(\mathbf{x}(T))/d\boldsymbol{\theta}$. For this, it suffices to augment the original dynamical system with its parameters[5] $\dot{\mathbf{z}} = [f(\mathbf{x}, \boldsymbol{\theta}), \mathbf{0}]$ and initial (later terminal) condition $\mathbf{z}(0) = [\mathbf{x}(0), \boldsymbol{\theta}]$. Defining the loss $\bar{L}(\mathbf{z}(T)) = L(\mathbf{x}(T))$, the adjoint of this augmented system is,

$$\mathbf{a}_{\mathbf{z}}(t) = \frac{d\bar{L}(\mathbf{z}(T))}{d\mathbf{z}(t)} = \left[ \frac{d\bar{L}(\mathbf{z}(T))}{d\mathbf{x}(t)}, \frac{d\bar{L}(\mathbf{z}(T))}{d\boldsymbol{\theta}} \right] = \left[ \frac{dL(\mathbf{x}(T))}{d\mathbf{x}(t)}, \frac{dL(\mathbf{x}(T))}{d\boldsymbol{\theta}} \right] = \left[ \mathbf{a}_{\mathbf{x}}(t), \frac{dL(\mathbf{x}(T))}{d\boldsymbol{\theta}} \right], \tag{E.1.7}$$

which contains the desired term $dL(\mathbf{x}(T))/d\boldsymbol{\theta}$. It now remains to describe the dynamics of the augmented system. By the same argument as for the state adjoint (E.1.5),

$$\frac{d\mathbf{a}_{\mathbf{z}}(t)}{dt} = \mathbf{a}_{\mathbf{z}}(t) \frac{d\dot{\mathbf{z}}}{d\mathbf{z}}. \tag{E.1.8}$$

By (E.1.7) and by unconcatenating $\mathbf{z}$,

$$= \left[ \mathbf{a}_{\mathbf{x}}(t), \frac{dL(\mathbf{x}(T))}{d\boldsymbol{\theta}} \right] \begin{bmatrix} \frac{df(\mathbf{x}(t))}{d\mathbf{x}(t)} & \frac{d\mathbf{x}(t)}{d\boldsymbol{\theta}} \\ \frac{d\mathbf{0}}{d\mathbf{x}(t)} & \frac{d\mathbf{0}}{d\boldsymbol{\theta}} \end{bmatrix} \tag{E.1.9}$$

$$= \left[ \mathbf{a}(t) \frac{df(\mathbf{x}(t))}{\mathbf{x}(t)}, \mathbf{a}(t) \frac{df(\mathbf{x}(t))}{d\boldsymbol{\theta}} \right]. \tag{E.1.10}$$

---

[5] Here $[\cdot, \cdot]$ denotes row concatenation.

Hence the following dynamical system can be evaluated,

$$\frac{d}{dt}\left[\mathbf{x}(t), \mathbf{a}(t), \frac{dL(\mathbf{x}(T))}{d\boldsymbol{\theta}}\right] = \left[f(\mathbf{x}(t), \boldsymbol{\theta}), \mathbf{a}(t)\frac{df(\mathbf{x}(t))}{\mathbf{x}(t)}, \mathbf{a}(t)\frac{df(\mathbf{x}(t))}{d\boldsymbol{\theta}}\right], \tag{E.1.11}$$

with terminal condition

$$\left[\mathbf{x}(T), \mathbf{a}(T), \frac{dL(\mathbf{x}(T))}{d\boldsymbol{\theta}}\right] = \left[\mathbf{x}(T), \frac{dL(\mathbf{x}(T))}{d\mathbf{x}(T)}, \mathbf{0}\right]. \tag{E.1.12}$$

Notice that to obtain the terminal condition, since it depends on $\mathbf{x}(T)$, the original dynamical system must be evaluated forward once.

### E.1.3 RNN adjoint

Let $\dot{\mathbf{x}} = f(\mathbf{x}, W) = W\phi(\mathbf{x}) - \mathbf{x} + B\mathbf{u}(t)$ and $L(\mathbf{x}(T)) = ||D\phi(\mathbf{x}(T)) - \mathbf{y}||^2$ for $y \in \mathbb{R}^d$. Furthermore, as it will be convenient, let $W = \sum_i^R \boldsymbol{\alpha}_i \otimes \boldsymbol{\beta}_i$. We will derive one by one the terms needed to characterize the parameter adjoint. First the Jacobian is

$$\frac{df(\mathbf{x}(t), \boldsymbol{\theta})}{d\mathbf{x}(t)} = \sum_i^R \boldsymbol{\alpha}_i \otimes (\boldsymbol{\beta}_i \odot \phi'(\mathbf{x}(t))) - I \tag{E.1.13}$$

where $\odot$ denotes the element-wise product, $I$ the identity matrix, and $\phi'$ the derivative of $\phi$. Next,

$$\frac{df(\mathbf{x}(t), \boldsymbol{\theta})_i}{dW_{jk}} = \begin{cases} 0 & i \neq j \\ \phi(\mathbf{x})_k & i = j \end{cases}, \tag{E.1.14}$$

that is

$$\frac{df(\mathbf{x}(t), \boldsymbol{\theta})}{dW} = I \otimes \phi(\mathbf{x}). \tag{E.1.15}$$

Finally the terminal condition,

$$\frac{dL(\mathbf{x}(T))}{d\mathbf{x}(T)_i} = \frac{d}{dx_i}\sum_j^d (D_j \cdot \phi(\mathbf{x}) - y_j)^2 = \sum_j^d (D_j \cdot \phi(\mathbf{x}) - y_j)(D_{ij}\phi'(\mathbf{x})_i) \tag{E.1.16}$$

that is,

$$\frac{dL(\mathbf{x}(T))}{d\mathbf{x}(T)} = \sum_j^d (D_j \cdot \phi(\mathbf{x}) - y_j)(D_j \odot \phi'(\mathbf{x})) = \phi'(\mathbf{x}) \odot \sum_j^d (D_j \cdot \phi(\mathbf{x}) - y_j)D_j \tag{E.1.17}$$

Or more explicitly, $\dot{\mathbf{a}}_{\mathbf{z}} =$

$$\begin{cases} \dot{\mathbf{a}}_{\mathbf{x}} = \left(\sum_i^R \boldsymbol{\alpha}_i \otimes \boldsymbol{\beta}_i \odot \phi'(\mathbf{x})\right)^T \mathbf{a}_{\mathbf{x}} - \mathbf{a}_{\mathbf{x}}, & \mathbf{a}_{\mathbf{x}}(T) = \phi'(\mathbf{x}(T)) \odot \sum_j^d (D_j \cdot \phi(\mathbf{x}(T)) - y_j)D_j \\ \dot{\mathbf{a}}_W = \mathbf{a}_{\mathbf{x}} \otimes \phi(\mathbf{x}), & \mathbf{a}_W(T) = 0 \end{cases}$$

$$\tag{E.1.18}$$

### E.2 Rank of the gradient

### E.2.1 The gradient as a composition of operators

In this section, we prove Theorems 1. In order to derive bounds on the singular values of $\nabla_W L = \mathbf{a}_W(0)$, we shall now consider $\mathbf{a}_{\mathbf{x}}$ and $\phi(\mathbf{x})$ as linear operators with integration. Namely,

$$\mathbf{a}_{\mathbf{x}}\mathbf{y} := \int_0^T \mathbf{a}_{\mathbf{x}}(t)y(t)dt \tag{E.2.1}$$

where $\mathbf{y} \in \mathcal{H}$ for $\mathcal{H}$ some suitable Hilbert space, such as $L^2$ for our case.

More formally, let $\mathbf{a}_{\mathbf{x}}, \phi(\mathbf{x}) \in \mathcal{B}_{0,0}$, where $\mathcal{B}_{0,0}$ is the Banach space of i) compact ii) bounded operators from $\mathcal{H}$ to $\mathbb{R}^n$, such that $\mathbf{a}_{\mathbf{x}}, \phi(\mathbf{x}) : \mathcal{H} \to \mathbb{R}^n$. In particular, we note that compactness follows from the image of $\mathbf{a}_{\mathbf{x}}$, that is $\mathbf{a}_{\mathbf{x}}(\mathcal{H})$, to be a vector subspace of $\mathbb{R}^n$ and therefore be of finite rank. By the same argument, $\phi(\mathbf{x})$ is compact, and therefore so is $\phi(\mathbf{x})^*$ by Schauder's theorem [55], where $^*$ is adjunction. Furthermore, notice that $\phi(\mathbf{x}), \mathbf{a}_{\mathbf{x}}$ are solutions of dynamical systems with differentiable right hand side and therefore bounded (in $\mathbb{R}^n$) if they are evaluated for finite time, and therefore bounded when seen as operators. The following result can now be applied:

**Lemma 2** ([55])**.** *Let $T \in \mathcal{B}_{0,0}$, then $T$ admits a singular value decomposition. Furthermore, this singular value decomposition is of finite rank.*

We can now prove the main theorem.

*Proof of Theorem 1.* Notice that the composition of the two operators is: $\mathbf{a_x} \circ \phi(\mathbf{x})^* = \nabla_W L$. Furthermore, the singular values of $\nabla_W L$ are,

$$\sigma_i^{\nabla_W L} = \min_{\substack{U \subseteq \mathbb{R}^n, \\ \dim U = \\ n-i-1}} \max_{\substack{\mathbf{y} \in U, \\ ||\mathbf{y}||=1}} ||\nabla_W L \mathbf{y}|| = \min_{\substack{U \subseteq \mathbb{R}^n, \\ \dim U = \\ n-i-1}} \max_{\substack{\mathbf{y} \in U, \\ ||\mathbf{y}||=1}} ||\mathbf{a_x}\phi(\mathbf{x})^*\mathbf{y}|| \tag{E.2.2}$$

which can be bounded as,

$$\min_{\substack{U \subseteq \mathbb{R}^n, \\ \dim U = \\ n-i-1}} \max_{\substack{\mathbf{y} \in U, \\ ||\mathbf{y}||=1}} ||\mathbf{a_x}\phi(\mathbf{x})^*\mathbf{y}|| \leq \min_{\substack{U \subseteq \mathbb{R}^n, \\ \dim U = \\ n-i-1}} \max_{\substack{\mathbf{y} \in U, \\ ||\mathbf{y}||=1}} ||\mathbf{a_x}||\,||\phi(\mathbf{x})^*\mathbf{y}|| \tag{E.2.3}$$

$$= \sigma_1^{\mathbf{a_x}} \min_{\substack{U \subseteq \mathbb{R}^n, \\ \dim U = \\ n-i-1}} \max_{\substack{\mathbf{y} \in U, \\ ||\mathbf{y}||=1}} ||\phi(\mathbf{x})^*\mathbf{y}|| \tag{E.2.4}$$

that is,

$$\tag{E.2.5}$$

$$\sigma_i^{\nabla_W L} \leq \sigma_1^{\mathbf{a_x}}\sigma_i^{\phi(\mathbf{x})^*} \tag{E.2.6}$$

Now notice that, for any operator, akin to the matrix case, $T_1, T_2$, the adjoint of their composition is $(T_1 T_2)^* = T_2^* T_1^*$. Furthermore,

$$\sigma_i^{(\nabla_W L)^T} = \min_{\substack{U \subseteq \mathbb{R}^n, \\ \dim U = \\ n-i-1}} \max_{\substack{\mathbf{y} \in U, \\ ||\mathbf{y}||=1}} ||(\mathbf{a_x}\phi(\mathbf{x})^*)^*\mathbf{y}|| = \min_{\substack{U \subseteq \mathbb{R}^n, \\ \dim U = \\ n-i-1}} \max_{\substack{\mathbf{y} \in U, \\ ||\mathbf{y}||=1}} ||\phi(\mathbf{x})\mathbf{a_x}^*\mathbf{y}|| \tag{E.2.7}$$

$$\leq \sigma_1^{\phi(\mathbf{x})} \min_{\substack{U \subseteq \mathbb{R}^n, \\ \dim U = \\ n-i-1}} \max_{\substack{\mathbf{y} \in U, \\ ||\mathbf{y}||=1}} ||\mathbf{a_x}^*\mathbf{y}|| \tag{E.2.8}$$

$$= \sigma_1^{\phi(\mathbf{x})}\sigma_i^{\mathbf{a_x}^*} \tag{E.2.9}$$

Noticing that $\sigma_i^{\mathbf{a_x}^*} = \sigma_i^{\mathbf{a_x}}$ and $\sigma_i^{\nabla_W L} = \sigma_i^{(\nabla_W L)^T}$,

$$\sigma_i^{\nabla_W L} \leq \sigma_1^{\phi(\mathbf{x})}\sigma_i^{\mathbf{a_x}} \tag{E.2.10}$$

Combining E.2.6 and E.2.10 we obtain the sought upper bound of Theorem 1,

$$\sigma_i^{\nabla_W L} \leq \min\left\{\sigma_1^{\mathbf{a_x}}\sigma_i^{\phi(\mathbf{x})}, \sigma_1^{\phi(\mathbf{x})}\sigma_i^{\mathbf{a_x}}\right\} \tag{E.2.11}$$

Similar steps can be used to derive the lower bound of Theorem 1, using instead the identity $\sigma_n^{T_1}||T_2\mathbf{y}|| \leq ||T_1 T_2 \mathbf{y}||$ where $T_1, T_2 \in \mathcal{B}_{0,0}$ and $\sigma_n^{T_1}$ is the smallest non-zero singular value of $T_1$. $\qquad\square$

We however note that, unlike the upper bound, the lower bound we provide does not have any numerical use, as the smallest singular value of the adjoint or of the firing rate is practically 0 (for example, well bellow machine precision).

We further point out that a more explicit characterization of the singular values of the gradient can be obtained. Let $U_i^{\mathbf{a_x}}, U_i^{\phi(\mathbf{x})}$ be the right singular vectors of respectively $\mathbf{a_x}$ and $\phi(\mathbf{x})$, so that $V_i^{\mathbf{a_x}}(t)$, $V_i^{\phi(\mathbf{x})}(t)$ are the left singular vectors and $\sigma_i^{\mathbf{a_x}}, \sigma_j^{\phi(\mathbf{x})}$ the singular values.

$$\nabla_W L = \sum_{i,j=1}^n \left(V_i^{\mathbf{a_x}} \otimes V_j^{\phi(\mathbf{x})}\right) \sigma_i^{\mathbf{a_x}}\sigma_j^{\phi(\mathbf{x})} \int_0^T U_i^{\mathbf{a_x}}(t)U_j^{\phi(\mathbf{x})}(t)dt \tag{E.2.12}$$

The characterization of the rank of the gradient has thus shifted to the firing rate and adjoint spaces. Furthermore, the integral is just the inner product between their right singular vectors and therefore of magnitude bounded by 1. Hence, for the dynamics of the weights to be large in a given direction in weight space $V_i^{\mathbf{a_x}} \otimes V_j^{\phi(\mathbf{x})}$, all three of: the singular values of the firing rate, the state adjoint, as well as their cofluctuation in time, must not be small.

### E.2.2 Weight gradient for time discretizations

Finally, we mention that our detour through functional analysis was for the sake of mathematical rigour, and that in practical applications, the RNN and its adjoint are evaluated at discrete time steps $0, \Delta t..., q\Delta t$. In that case, the gradient can be estimated as a simple matrix-matrix product. Let $A = [\mathbf{a_x}(0), ..., \mathbf{a_x}(q\Delta t)]$ and $B = [\phi(\mathbf{x})(0), ..., \phi(\mathbf{x})(q\Delta t)]$. Then $\nabla_W L = \Delta t A B^T$, and the bounds $E.2.6$ and $E.2.10$ follow from classic matrix-matrix product bounds [56]. In particular, given that $\phi(\mathbf{x})$ and $\mathbf{a_x}$ are smooth, without loss of generality,

$$\sigma_i^{\nabla_W L} = \lim_{\min_j(t_{j+1}-t_j)\to 0} \sigma_i^{AB^T} \tag{E.2.13}$$

This simple matrix-matrix product opens up the possibility of fast RNN adjoint implementations as, unlike computing $\mathbf{a_x}\frac{df}{dW}$ in general adjoint solvers, which requires $O(qn^3)$ time and $O(n^2)$ memory complexity for an $n$-dimensional RNN and $q$ time steps, here the time complexity drops to $O(qn^2)$.

### E.2.3 Rank of linear RNNs

In this section we prove Theorem 2.

*Proof of claim 1.* If $W = \sum_i^R \boldsymbol{\alpha}_i \otimes \boldsymbol{\beta}_i$, then by Lemma 1, the dynamics of the adjoint are constrained to the span of the rows of $W$, namely, $\dot{\mathbf{a}}_\mathbf{x} \in \text{span}\{\boldsymbol{\beta_i}\}$. Therefore, $\mathbf{a_x} \in \text{span}\{\boldsymbol{\beta_i}\} \cup \{\mathbf{a_x}(T)\}$, which is a at most $R + 1$ dimensional subspace. If $\mathbf{a_x}$ is constrained to a at most $R + 1$ dimensional subspace, then $\dot{\mathbf{a}}_W = \mathbf{a_x} \otimes \mathbf{x}$ is also constrained to a at most $R + 1$ dimensional subspace. Since $\mathbf{a}_W(T) = 0$, $\mathbf{a}_W$ is constrained to the same subspace as its dynamics, and in particular, $\text{rank}\,\mathbf{a}_W(0) = \text{rank}\,\nabla_W L^{(0)} \leq R + 1$. $\square$

*Proof of claim 2.* Suppose $W^{(k)} = \sum_i^{2R+m+d} \boldsymbol{\alpha}_i^{(k)} \otimes \boldsymbol{\beta}_j^{(k)}$, where $\boldsymbol{\alpha}_i^{(k)}, \boldsymbol{\beta}_i^{(k)} \in \text{span}\{\boldsymbol{\alpha}_i^{(0)}\} \cup \{\boldsymbol{\beta}_i^{(0)}\} \cup \{B_i\} \cup \{D_i\} := V$. In particular, notice that $V$ is only dependent on the initial weight. Then by a similar argument as above, $\mathbf{a_x}^{(k)} \in V$, which implies $\mathbf{a}_W(0)^{(k)} = \nabla_W L^{(k)} \in V$. Therefore $W^{(k+1)} = W^{(k)} + \gamma \nabla_W L^{(k)} = \sum_i^{2R+m+d} \boldsymbol{\alpha}_i^{(k+1)} \otimes \boldsymbol{\beta}_j^{(k+1)}$ with $\boldsymbol{\alpha}_i^{(k+1)} \in V$. That is $\text{rank}\,W^{(k+1)} \leq 2R + m + d$. $\square$

*Proof of claim 3.* Mutatis mutandis $\boldsymbol{\beta}^{(k+1)} \in V$, that is $\mathbf{x} \in V$. Therefore, $\mathbf{W}^{(k)} = \sum_{ij}^{2R+m+d} c_{ij}^{(k)} \boldsymbol{\alpha}_i \otimes \boldsymbol{\alpha}_j$ for some $c_{ij}^{(k)}$'s, where $\boldsymbol{\alpha}_i \in V$. Or equivalently, $\mathbf{W} = \sum_{ij}^{2R+m+d} \boldsymbol{\alpha}_i \otimes \boldsymbol{\alpha}_j \otimes \mathbf{c}_{ij}$. That is, $\text{rank}\,\mathbf{W} \leq (2R + m + d)^2$. $\square$

### E.3 Extensions of our results

**Loss integrated over time**. Commonly, the loss considered might be integrated,

$$\mathcal{L}(T) := \int_0^T L(\mathbf{x}(t), \mathbf{y}(t)) dt \tag{E.3.1}$$

The parameter adjoint is dependent only linearly on the state adjoint, we may therefore integrate the state adjoint for all initial conditions.

$$\mathcal{L}(T) = \left( \int_T^0 \mathbf{a_x}(t) + \int_t^0 \dot{\mathbf{a}}_\mathbf{x}(t')dt' \right) dt \tag{E.3.2}$$

The term inside the first integral is just the solution of time-varying autonomous LDS, therefore,

$$\int_T^0 \mathbf{a_x}(t) dt = \int_T^0 \Phi(0, t) \frac{dL(\mathbf{x}(t), \mathbf{y}(t))}{d\mathbf{x}(t)} dt \qquad (E.3.3)$$

Where $\Phi$ is the linear dynamical system state transition matrix [57]. But notice that this is the solution to the controlled LDS,

$$\dot{\mathbf{a}}_\mathbf{x} = \mathbf{a_x} \frac{df(\mathbf{x}(t))}{d\mathbf{x}(t)} + \frac{dL(\mathbf{x}(t), \mathbf{y}(t))}{d\mathbf{x}(t)}, \quad \dot{\mathbf{a}}_\mathbf{x}(T) = 0 \qquad (E.3.4)$$

In the specific case of an RNN,

$$\dot{\mathbf{a}}_\mathbf{x} = \left( \sum_i^R \alpha_i \otimes \beta_i \odot \phi'(\mathbf{x}) \right)^T \mathbf{a_x} - \mathbf{a_x} + \phi'(\mathbf{x}(t)) \odot \sum_j^d (D_j \cdot \phi(\mathbf{x}(t)) - \mathbf{y}_j) D_j, \quad \dot{\mathbf{a}}_\mathbf{x}(T) = 0$$
$$(E.3.5)$$

Therefore, Theorem 1 remains unchanged for a loss integrated over time. For Theorem 2, Claims 2-3 remain unchanged, while Claim 1 becomes $\mathrm{rank}\, \nabla_W L^{(0)} \leq R + \min\{d, m\}$.

**Gradient with respect to other parameters**. So far we have focused on the gradient of the weights of the RNN. As we have seen, the space over which the state adjoint $\mathbf{a_x}(t)$ evolves as well as the trajectories of the system itself $\mathbf{x}(t)$ determine the space over which the gradient evolves. But those are respectively dependent on the decoder $D$ and encoder $B$ of the system. If all parameters $D, B, W$ of the system are optimized simultaneously, as is most often the case, we may wonder how our bounds hold.

First, for $B$, notice that by a similar argument as for $W$, $\frac{df(\mathbf{x}, B\mathbf{u}(t))}{dB} = I \otimes \mathbf{u}(t)$, so that $\dot{\mathbf{a}}_B = \mathbf{a_x} \otimes \mathbf{u}(t)$. Therefore, following essentially the same derivation as that of Theorem 1, the following bound can be derived,

$$\sigma_i^B \leq \min\{\sigma_1^{\mathbf{a_x}} \sigma_i^\mathbf{u}, \sigma_1^\mathbf{u} \sigma_i^{\mathbf{a_x}}\}. \qquad (E.3.6)$$

By a similar argument as for the derivation of Theorem 2, $B_i^{(k)} \in \mathrm{span}\{\alpha_i\} \cup \{\beta_i\} \cup \{B_i\} \cup \{D_i\}$.

Second, for $D$, since $\frac{df}{dD} = 0$, that is $\dot{\mathbf{a}}_D = 0$,

$$\mathbf{a}_D(T) = \mathbf{a}_D(0) = \nabla_D L = \frac{dL(T)}{dD}. \qquad (E.3.7)$$

In other words, the gradient of the loss with respect to the decoder weights have zero dynamics.

The gradient of a functional w.r.t. a given parameter is independent of the gradient of that functional with respect to another parameter if these two parameters do not depend on one another. Therefore these results hold regardless of which combination of $W$, $B$ or $D$ is optimized.

**Batched updates**. Most often, the weights are updated in batches. That is $\Delta W^{(k)} = Q^{-1} \sum_q^Q \nabla_W L^{(k,q)}$ where $q$ is the index over the batched dimension. Since the column and row spaces of $\nabla_W L^{(k,q)}$ remain unchanged, Theorem 2 2-3. remain unchanged, while 1. becomes $\mathrm{rank}\, \Delta W^{(0)} \leq R + \min\{d, m\}$. For Theorem 1, the common singular value identities [56] $\sigma_{i+j-1}(A + B) \leq \sigma_i(A) + \sigma_j(B)$ and $\sigma_i(cA) = c\sigma_i(A)$ for $c \in \mathbb{R}_0^+$ can be used. Then, $\sigma_{\sum i_q - Q + 1}(\Delta W^{(k)}) \leq Q^{-1} \sum_q^Q \sigma_{i_q}(\nabla_W L^{(k,q)})$.

**Momentum-based optimizers**. Momentum-based optimizers such as Adam [53] are commonly used to train RNNs on behavioural tasks. Here we focus on the first moment, a similar derivation can be undertaken for higher moments. In that case, a momentum variable is introduced, which is updated as $M^{(k+1)} = \beta M^{(k)} + (1 - \beta)\nabla_W L^{(k)}$, where $\beta$ determines the speed of the exponential decay. The weights are then updated as $W^{(k+1)} = W^{(k)} - \alpha M^{(k+1)}$, where $\alpha$ is the learning rate. Which implies, $\Delta W^{(k)} = W^{(k+1)} - W^{(k)} = -\alpha \sum_j^k (1 - \beta)^j \nabla_W L^{(j)}$. Using the same identities are for batched updates, $\sigma_{\sum i_q - k + 1}(\Delta W^{(k)}) \leq \sum_j^k (1 - \beta)^j \sigma_{i_q}(\nabla_W L^{(j)})$.

## E.4 Numerical simulations

Similarly to [14], we illustrate our mathematical results on random RNNs. Since constant inputs are one dimensional ($B\mathbf{u} = \sum B_i u_i$), we instead use time-varying inputs parameterized with LDS:

$$\dot{\mathbf{u}} = M\mathbf{u} - \mathbf{u} \quad \mathbf{u}(0) = \mathbf{u}_0 \tag{E.4.1}$$

where $\mathbf{u}(t) \in \mathbb{R}^m$, $M_{ij} \sim \mathcal{N}(0, 1\sqrt{m})$, $\mathbf{u}_0 \sim \mathcal{N}(0, \mathbf{1}/2)$. The target outputs are set as $\mathbf{y} \in \mathbb{R}^l$, $\mathbf{y}_i \sim \mathcal{U}(-1, 1)$. The loss is defined as,

$$L(W) = ||D\phi(\mathbf{x})(T) - \mathbf{y}||^2 \tag{E.4.2}$$

where $\mathbf{x}$ is the solution of an RNN as considered thus far.

In Sup. Fig. 7 we show the effect of varying $\phi$, the rank $R$ of the initial weights as well as the standard deviation (or strength) $g$ of the initial weights such that $W_{ij}^{(0)} \sim \mathcal{N}(0, g^2)$.

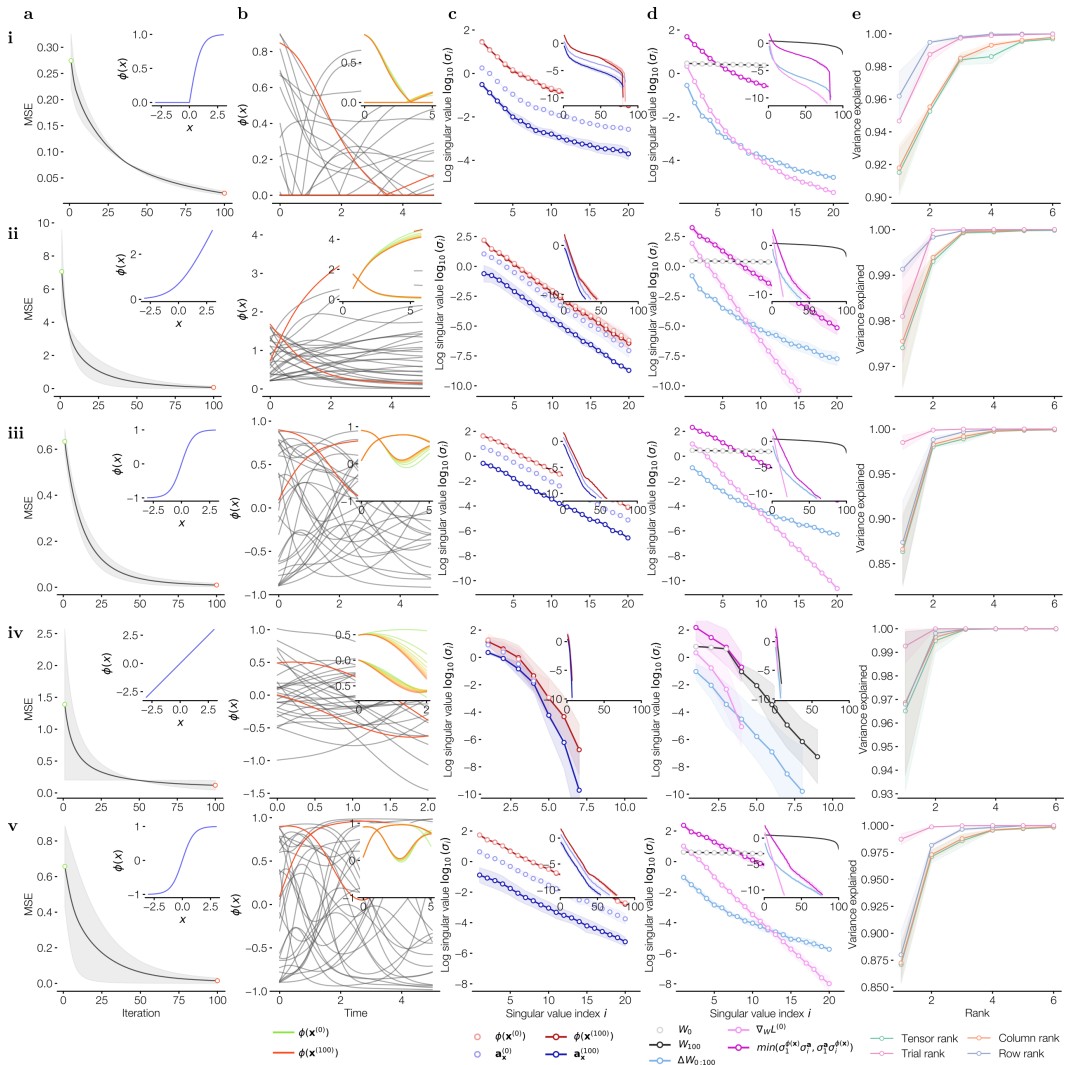

Supplementary Figure 7: **Singular values of adjoints and gradients**. **a**. Loss over training. Inset: activation function. **b**. Activity during the last trial (black). Inset: activity of two example neurons over training. **c**. Singular values of the firing rate and adjoint. Inset: additional singular values. **d**. Singular values of the gradient, weights, and the bound we derive. **e**. Variance explained per rank of the tensor decomposition of weight tensor $\mathbf{W} - W^{(0)} \otimes \mathbf{1}$. We additionally plot the variance explained over performing matrix decomposition on all possible unfoldings of the weight tensor. **Architectures**. Unless noted, the initial weight std was $g = 1.5$, the input and output dimensions $m = d = 2$. **i**. Rectified tanh. We found that non-smooth activation functions seem to give the slowest decay of the singular values of the firing rate and adjoint. **ii-iii**. Softplus and tanh, which are the most common activation functions in neuroscience, have exponentially decaying firing rate and adjoint singular values, and therefore (by Theorem 1) exponentially decaying gradient singular values. **iv**. Low rank ($R = 3$) linear RNN. As per Theorem 2, the first gradient step is of rank $R + 1 = 4$. **v**. Tanh in a chaotic regime ($g = 2.1$). Despite being in a chaotic regime, the firing rate, the adjoint, and therefore (by Theorem 1) the gradient have exponentially decaying singular values.

