# OpenReview forum: "Low Tensor Rank Learning of Neural Dynamics"
_NeurIPS.cc/2023/Conference — NeurIPS 2023 poster_

### Official Review · Reviewer_Qcae · 2023-07-03

**Soundness:** 3 good
**Presentation:** 3 good
**Contribution:** 3 good
**Rating:** 7
**Confidence:** 2

**Summary:**

Proposes a low tensor rank recurrent neural network (ltrRNN) architecture, in which the tensor constructed by stacking the RNN weight matrices of different trials is constrained to have low tensor rank. Empirically shows that ltrRNNs can fit neural recordings during a motor learning task, achieving lower unexplained variance than baseline methods. Then, demonstrates that an RNN trained to perform the same motor learning task yields dynamics that can also be fit well with ltrRNNs (i.e. low tensor rank dynamics). Lastly, theoretically analyzes how gradient-based optimization can lead to low rank, establishing upper bounds on matrix and tensor ranks for RNNs.

Discloser: The research area of the current paper (interplay between machine learning and neuroscience) falls outside my expertise, and so it is difficult for me to assess the novelty and significance of some of the contributions. My review mainly focuses on presentation, soundness, and the theoretical analysis of gradient descent constraining tensor rank of weight updates.


**Strengths:**

1. Reads relatively well.

2. As far as I am aware, the technique proposed for modeling structure in a task learning process through low tensor rank of weights across different iterations is novel. I found the idea of examining such low rank structure insightful --- existing characterizations of (low rank) structure during gradient-based learning typically focus on weights/representations of a single iteration. I believe this concept may turn out useful for future study of implicit regularization of gradient descent.

3. Analyzing the dynamics of gradient-based optimization is a subject of significant interest in recent years. The bounds on the matrix and tensor ranks of the gradient and weights of a continuous RNN contribute to a line of works suggesting that gradient descent leads to low rank solutions (e.g. in matrix and tensor factorizations as well as non-linear networks [1, 2, 3, 4, 5]). With that said, the form of the RNN considered is unorthodox in terms of its update rule and in being continuous-time, which may limit the impact of these results.

[1] Li, Zhiyuan, Yuping Luo, and Kaifeng Lyu. "Towards resolving the implicit bias of gradient descent for matrix factorization: Greedy low-rank learning." arXiv preprint arXiv:2012.09839 (2020).

[2] Razin, Noam, Asaf Maman, and Nadav Cohen. "Implicit regularization in tensor factorization." International Conference on Machine Learning. PMLR, 2021.

[3] Razin, Noam, Asaf Maman, and Nadav Cohen. "Implicit regularization in hierarchical tensor factorization and deep convolutional neural networks." International Conference on Machine Learning. PMLR, 2022.

[4] Boursier, Etienne, Loucas Pillaud-Vivien, and Nicolas Flammarion. "Gradient flow dynamics of shallow relu networks for square loss and orthogonal inputs." Advances in Neural Information Processing Systems 35 (2022): 20105-20118.

[5] Timor, Nadav, Gal Vardi, and Ohad Shamir. "Implicit regularization towards rank minimization in relu networks." International Conference on Algorithmic Learning Theory. PMLR, 2023.


**Weaknesses:**

I found the empirical evidence to be somewhat unsatisfactory since it only includes two datasets. Experiments on further tasks/datasets may greatly solidify the viability of ltrRNN. Such experiments can reveal whether the low tensor rank dynamics are specific to the type of neural recording data examined here or it is more general (and thus significant).

An additional (more minor) comment: Some of the terms used are non-standard in the machine learning literature. Since results such as those in Section 6 can be of interest to researchers not familiar with this terminology it may be best to clarify (e.g. in footnotes or an appendix) their meaning. For example, the terms “trial”, “task condition”, “chaotic regime”.

**Questions:**

Have you applied the ltrRNN to other datasets? Or is the purpose of the architecture specific to the kind of data reported in the paper?

**Limitations:**

The authors have adequately addressed possible limitations of their work.

---

> ### Author Rebuttal · Authors · 2023-08-10
>
> We thank the reviewer for the positive feedback regarding the novelty and utility of our work.
>
> **Q1. Relationship between continuous-time and discrete-time RNNs.**
>
> **A.**  We used a formulation of a continuous-time RNN that is commonly used in neuroscience applications of machine learning research [1-3], as these equations can be seen as approximations to more biologically motivated systems of ODEs describing neural circuit dynamics [4].  More generally, there has been a recent surge of interest in applications of ODEs to machine learning [5], as moving to the continuous domain enables the large body of literature on the mathematical theory behind ODEs and dynamical systems to be leveraged. For example, we exploited the adjoint state method to derive our analytical bounds on the ranks of the weight updates. In broad strokes, the adjoint method works by defining a new ("adjoint") dynamical system that is driven by the time-inverted dynamics of the original ODE (see Supplementary Materials D.1.1-2 for a precise mathematical treatment). However, this is not always possible for a discrete-time RNN ($\mathbf{x}_{t+1} = f(\mathbf{x}_t,\theta)$), as a non-invertible $f$ would mean that the dynamics of $\mathbf{x}_t$ cannot be time-inverted (as opposed to the continuous-time RNN, $\mathbf{\dot x} = f(\mathbf{x},\theta)$). Therefore, while we expect that an extension of our analyses from continuous  to discrete-time RNNs is possible, it would be a non-trivial exercise.
>
> Regarding the update rule, we note that while our bounds in the main text are based on gradient descent, they can be extended to Adam (line 777 of Supplementary Material D). In the revised version of the manuscript, we will point the reader to the relevant section of the Supplementary Material.
>
> Finally, we thank the reviewer for pointing out these additional citations on implicit regularization, which we had missed. We will cite these in the discussion section.
>
> **Q2. Unfamiliar terminology.**
>
> **A.** We thank the reviewer for pointing out the neuroscience-specific jargon. We will modify the revised version with the following definitions: Typically, in a neuroscience experiment, the animal is required to perform the same task over many repetitions in order to quantify the variance of neural activity. Each of these repetitions is a *trial*. In addition, a typical task often has many "conditions" that change the target output of the task (for RNNs different condition corresponds to mapping a different input to another target output). Regarding chaotic dynamics, it has been shown that in RNNs with weights $\sim \mathcal{N}(0,\sigma^2)$, increasing $\sigma^2$ leads to a transition from non-chaotic dynamics (the system has non-positive Lyapunov exponents and $\bf x$ settles into a fixed point attractor) to chaotic dynamics (the system has at least one positive Lyapunov exponent and $\bf x$ displays large fluctuations over time, see e.g. [3]). We will incorporate these definitions in the introduction and in footnotes in the revised version.
>
> **Q3. Application to other datasets.**
>
> **A.** To address the reviewer's question, and to demonstrate that the idea of low-tensor-rank learning is not specific to motor learning, we have now performed additional experiments (see General Response to Reviewers for details) which confirm that other forms of learning in neural data and in task-trained RNNs are often low rank.
>
> In general, however, we acknowledge that we do not expect *all* changes in brain activity to necessarily have low tensor rank weights. An important counterexample is the bump attractor network [9]. In this model, each neuron has excitatory (positive weight) connections with its adjacent neurons, and inhibitory (negative weight) connections with distal neurons, e.g. $W_{ij}^{(k)}=w_E$ if $|i-j|<r$, else $W_{ij}^{(k)} = -w_I$, where $w_E,w_I>0$ and $r$ is the radius of the excitatory connections. Because $W^{(k)}$ has banded structure along its diagonal, it will have full matrix rank. Therefore in the case of the brain learning a bump attractor network, and since a high matrix rank in one of the slices of the tensor implies a high tensor rank, we expect $\bf W$ to have high tensor rank. As this is an important counterexample to the low tensor rank learning framework, we will incorporate it into the discussion of our revised paper regarding universality and limitations of the ltrRNN framework.
>
> **References**
>
> [1] Turner, Dabholkar, and Barak. "Charting and navigating the space of solutions for recurrent neural networks." *NeurIPS* 2021.
>
> [2] Schuessler, Mastrogiuseppe, Dubreuil, Ostojic, and Barak. "The interplay between randomness and structure during learning in RNNs." *NeurIPS* 2020.
>
> [3] Kadmon, Timcheck, and Ganguli. "Predictive coding in balanced neural networks with noise, chaos and delays." *NeurIPS* 2020.
>
> [4] Dayan and Abbott. Theoretical Neuroscience. *MIT Press*, 2001.
>
> [5] Chen, Rubanova, Bettencourt, and Duvenaud. "Neural ordinary differential equations." *NeurIPS* 2018.
>
> [6] Pontryagin, Mishchenko, Boltyanskii, and Gamkrelidze. The mathematical theory of optimal processes. *Classics of Soviet mathematics*, 1962.
>
> [7] Miller and Fumarola. "Mathematical equivalence of two common forms of firing-rate models of neural networks." *Neural Computation*, 2012.
>
> [8] Humphreys, Daie, Svoboda, Botvinick, and Lillicrap. "BCI learning phenomena can be explained by gradient-based optimization." *bioRxiv* 2022. https://doi.org/10.1101/2022.12.08.519453
>
> [9] Compte, Brunel, Goldman-Rakic, and Wang. "Synaptic mechanisms and network dynamics underlying spatial working memory in a cortical network model." *Cerebral Cortex* 2000.

---

> > ### Comment · Reviewer_Qcae · 2023-08-11
> >
> > Thank you for the detailed response, I have read it and the other reviews carefully.
> >
> > In light of the additional experiments, structural changes to presentation, and additional clarifications delineated in the response, I am raising my initial recommendation to accept.

---

### Official Review · Reviewer_jwhe · 2023-07-04

**Soundness:** 2 fair
**Presentation:** 2 fair
**Contribution:** 2 fair
**Rating:** 4
**Confidence:** 1

**Summary:**

The presented work investegated the 3-tensor formed by the weight matrices of RNNs across trials and found it is low-rank. The authors also conducted a mathematical proof that the weights learned by gradient-descent on low-dimensioanl tasks are low-rank.

**Strengths:**

First I should ackonwledge that I am not an expert in neural science, and I found it hard to fully understand this paper, so I can only provide limited insights.

Strengths:
- The low-rank property of RNN is interesting and is potentially useful for undertanding the neural dynamics and develop neural network architechures.
- The mathematical framework could be valuable for comprehendingnature of gradient descent.

------

The author rebuttal has addressed most of my questions. However, since I'm not familiar with neural science, I decide to keep my current rating to this paper as borderline reject with the lowest confidence value.

**Weaknesses:**

- The empirical results seems to be limited since the experiments are only performed on motor learning tasks. It's unclear how universal the low-rank property is.

**Questions:**

It might be my problem but I found this paper hard to understand. For example, what does "weights change smoothly over trials" (line 129) mean? I suppose trials are unordered, so what does smoothness mean here? Also, what does "smooth covariance matrix" mean in line 132? $A$ is the covariance matrix of which variable? In the "slow timescale variability in data" (line 136), the variability of what? In "we compared the performance of the full tensor RNN to a static RNN", what is a static RNN? It would be beneficial if the authors provided clear mathematical definitions for both the model and the task, as I currently feel somewhat out of context.

---

> ### Author Rebuttal · Authors · 2023-08-10
>
> We thank the reviewer for noting that our framework is valuable for understanding gradient based learning as well as neural dynamics.
>
> **Q1. Generality of low-rank learning dynamics.**
>
> **A.** We have now included three additional task-trained RNN simulations and an additional neural dataset to validate the generality of our results. See Qcae Q4 and General Response to Reviewers for additional details.
>
> **Q2. Lack of clarity for non-specialist readers.**
>
> **A.**  We appreciate the reviewer's feedback that our paper may be unclear to readers without expertise in neuroscience. As our work intersects machine learning and computational neuroscience, we believe it is important that their presentation can be understood by both communities. We will therefore clarify the neural data application section of the manuscript by better defining terminology that is not standard in machine learning. We will also highlight earlier in the manuscript the mathematical results and their associated numerical simulations in order to cue potential readers who may be interested in gradient learning dynamics, and will additionally provide better insight regarding their broader significance in machine learning research.
>
> **Q3. Trial ordering.**
>
> **A.** We thank the reviewer for pointing out the lack of clarity of the smoothness constraint we impose on the weights as well as about its relationship to the concept of a *trial*.
> One trial corresponds to one repetition of the task of the animal: in Figure 1, this entails moving the cursor from the center of the screen to the target. The different targets in the experiment represent different task conditions. Trials are indeed ordered in the sense that trial $k+1$ occurs after trial $k$ in the original experiment. However each trial is assigned a random task condition.
>
> **Q4. Smooth changes in weights over trials.**
>
> **A.** Over learning, neural activity changes in order to perform the computations necessary for the task. This change in neural activity is generally accepted to be due to changes in the synaptic weights, which evolve over slow timescales due to well-known plasticity mechanisms such as Hebbian learning [1-2]. Because synaptic plasticity is slower than stimulus-driven changes in neural firing, we assume a separation of timescales such that the weight matrix stays constant within a trial but changes from one trial to the next (this separation of timescales has also previously been emphasized within neuroscience [3-4]). This is parametrized by defining the weight matrix on trial $j$ as $W^{(j)} \in \mathbb{R}^N$. When we say that the weights change smoothly over trials (e.g., in line 129), we mean that $W^{(j+1)}-W^{(j)}$ should be small.
>
> **Q5. Smooth covariance matrix.**
>
> **A.** As a reminder, on trial $j$ the weight matrix is $W^{(j)} = \sum_r^R c_r^{(j)} {\mathbf a}_r \otimes {\mathbf b}_r$, where $c_r^{(j)}$ is the $j$th element of $\mathbf{c}_r$. Therefore the trial factors $\mathbf{c}_r$ represent how $W^{(j)}$ changes over trials. Our assumption that the $W^{(j)}$ change smoothly over trials thus corresponds to an assumption that the $c_r^{(j)}$ changes smoothly over $j$.
>
> More practically, we implement smoothness of the weights over trials by constraining the temporal covariance between $W^{(i)}$ and $W^{(j)}$. This can be done by first stacking the trial factors into a matrix $\mathbf{c}=[\mathbf{c}_1, ..., \mathbf{c}_r]$, and by assuming that its covariance matrix of $Cov(\mathbf{c}) \in \mathbb{R}^{K \times K}$ is given by a smooth kernel. In section 2 of the main text and Supplementary Material A we detail how this can be achieved. We will reword this section of the revised paper to clarify this smoothness constraint for the reader.
>
> **Q6. Slow timescale variability in data.**
>
> **A.** In line 136 we refer to the variability of the neural activity of the recording. By slow timescale, we mean changes from trial to trial (i.e., over minutes or even days), rather than rapid changes (i.e., over milliseconds). Recent work has suggested that this separation of timescales captures different biological processes, with rapid changes representing processing of external stimuli and slower changes representing learning [3]. Here we further assume that the variability of neural activity over trials due to learning can be accounted for by changes in $W^{(j)}$, while the variability of neural activity within a single trial is the result of the ODE defining the network for a fixed $W^{(j)}$ (see also our response to Q4).
>
> **Q7. Static RNN.**
>
> **A.** By "static RNN" (line 160) we mean an RNN whose weights remain fixed between trials (i.e., $W^{(i)} = W^{(j)}$ for all $i,j$). That is, we fitted a single set of weights ($N^2$ parameters) for the entire neural recording. We will now refer to this model simply as "an RNN with fixed weights over trials".
>
> **Q8. Mathematical definitions of the model.**
>
> **A.** We agree with the reviewer's general point that having a good grasp of the ltrRNN model and training procedure is important for the reader to understand the rest of the paper. We had originally provided a detailed definition of the model and training procedure in Supplementary Material A while the main text only provided a high-level description. In the revised manuscript we will i) provide a short description of the training procedure in the main text, ii) systematically point at the relevant section in supplementary material iii) provide pseudocode of the ltrRNN fitting procedure.
>
> **References**
>
> [1] Confavreux, Basile, et al. "A meta-learning approach to (re) discover plasticity rules that carve a desired function into a neural network." *NeurIPS* 2021.
>
> [2] Stevenson, Ian, and Konrad Koerding. "Inferring spike-timing-dependent plasticity from spike train data." *NeurIPS* 2011.
>
> [3] Soulat, Hugo, et al. "Probabilistic tensor decomposition of neural population spiking activity." *NeurIPS* 2021.

---

> ### Comment · Reviewer_jwhe · 2023-08-15
>
> Thank the authors for their responce and clarification. Since I'm not familiar with neural science, I decide to keep my current rating to this paper as borderline reject with the lowest confidence value. I encourage the authors to add more background introduction and notation definitions in the future revision.

---

### Official Review · Reviewer_T6eW · 2023-07-05

**Soundness:** 3 good
**Presentation:** 3 good
**Contribution:** 3 good
**Rating:** 7
**Confidence:** 4

**Summary:**

The work "Low Tensor Rank Learning of Neural Dynamics" investigate the low-rankness of RNNs with application to neural data, i.e. neural signals of a test subject performing a motor task.
The authors describe that RNNs are of low rank in the trial mode when parametrized as a 3-tensor where one dimension represents the different trials. The findings are validated by showing that a low-rank parametrized RNN is able to fit the motor task with similar accuracy to the full-rank network.
In their theory section, the authors propose two theorems that show the boundedness of the singular values of RNN weight matrices.

**Strengths:**


This is a solid paper. Particularly interesting is:

- Investigation of low-ranked RNNS as a model for real-world data - in this case neural signals for a motor task.
- Analysis of the gradient dynamics, i.e. the singular values of RNN gradients,
- Extensive numerical tests to validate the propositions, supplemented by code examples.
- Comprehensive Related work section, which is important for such interdisciplinary work.

**Weaknesses:**


- Some method details for the ltrRNN training need clarification (see questions) In particular, an algorithm or some more mathematical details on tensor format and weight updates are required.
This is the major drawback on this papers presentation, in my opinion.
- The computational cost of training and used hardware should be described.


**Questions:**



- Line 97: Please clarify: Do you ask if the weight tensor itself is low-rank or if the updates are low-rank?

- Line 118: Add a reference or proof (in the appendix at least) for the statement. You say to yourself that this is a non-trivial statement.

- Line 96: please explain the meaning of x and u in the context of your application.

- Line 168: It is not clear, what you mean by your method. I suppose, that you mean the formulation of an order three tensor of low-rank structure as given in the Eq. of Line 95. If so, it is unclear, how the tensor is represented (Tucker format, Tensor-Trains,...) and how the network is trained. Vanilla on the factors, or with dynamical low-rank methods. Can you comment on this? An algorithm, reference, or some equations to explain the method would be necessary.

- Sec4:  The authors need to specify
	* a) which low-rank tensor format is chosen to save the weight tensors
	* b) which update method / low-rank integrator or optimizer is used to compute the weight updates

- Line 168: You say that your method outperforms truncated SVD. My question is, how is truncated SVD training applied? Is a full rank update and afterward a truncated SVD performed in each training iteration, or do you do something else?

- Line 182: "Compared to PCA on neural data, ltrRNNs yield more interpretable visualizations..." First, (minor comment), there is a typo "intepretable". Second, this is an interesting aspect. I think, that training an RNN instead of direct application of PCA means, that you have first a differentiable (and thus smooth) model representation of the neural data, which is then, of course, nicer interpretable, and more visualizable. My questions:
	1. Are the neural data smoothed out in some sense, before applying PCA, is the smoothness of Fig.3d a result of the plotting tool or is the neural data smooth, i.e. without large jumps or discontinuities?
	2. How comes, that the neural data seem to be somewhat chaotic?
I am by no means an expert in this application field, but find this intriguing.


- Line 225 and the following: You describe how ltrRNN uncovers the low-rank structure. As per Appendix A, it seems that you construct the ltrRNN architecture such that it is low-rank per definition.  Thus the network has now another chance to learn low-rank features. Can you comment on this for clarification?

- Line 256:  One should also mention [2] as one of the fundamental works of adjoint-based automatic differentiation.



[2] Griewank, Andreas, Walther, Andrea; Introduction to Automatic Differentiation; PAMM; https://doi.org/10.1002/pamm.200310012


**Limitations:**

The authors have described various limitations of their work, and proposed concepts on how to deal with them in future work.

---

> ### Author Rebuttal · Authors · 2023-08-10
>
> We thank the reviewer for the positive feedback regarding our paper and its potential reach as interdisciplinary work.
>
> **Q1. Clarification of ltrRNN model and training.**
>
> **A.** We agree with the reviewer that having a good grasp of the ltrRNN training procedure is important for the reader to understand the rest of the paper. We had originally provided a detailed definition of the training procedure in Supplementary Material A. In the revised manuscript we will i) provide a short description of the training procedure in the main text, ii) systematically point at the relevant section in Supplementary Material, and iii) provide pseudocode of the ltrRNN fitting procedure.
>
> **Q2. Computational cost of training.**
>
> **A.** We agree that it is useful for the reader to know the computational cost of the method. Supplementary Material A currently includes a table describing the time necessary to fit ltrRNNs of different numbers of neurons (given that this is the main bottleneck) to the neural data application in Figure 1. In the revised manuscript we will point to this table directly when describing the ltrRNN model.
>
> **Q3. Low rank weight tensor or low rank weight updates.**
>
> **A.** We thank the reviewer for pointing out the lack of clarity in line 97 regarding whether the low-rank hypothesis applies to the weight tensor $\bf W$ or the tensor $\bf \Delta W $ that stacks the weight update matrices: $\Delta W^{(k)} = W^{(k)} - W^{(k-1)}$. We indeed mean that $\bf W$ itself is low tensor rank. We will correct this wording in the new version to read: "Here, we ask whether the weights have low tensor rank when they are updated over the course of learning".
>
> **Q4. Reference for the eigenvalue equation in line 118.**
>
> **A.** We thank the reviewer for noticing this overlook. The derivation is already present in Supplementary Material A.3, but we neglected to mention it in the main text. We will add a pointer to the derivation to the revised version.
>
> **Q5. Meaning of x and u.**
>
> **A.** RNNs of the form in line 96 are commonly used in neuroscience as models of networks of biological neurons [1,2], where the state $\bf x$ of the RNN models the membrane potential of the neurons, $\phi(\mathbf{x})$ models their firing rate (i.e. activity), and $\bf u$ represents input activity from other brain regions to the neurons. We will incorporate these definitions into the model description of the main text to aid the reader.
>
> **Q6.  Weight tensor representation and training.**
>
> **A.** We agree with the reviewer that the phrasing of line 168 is misleading. By *our method* we mean ltrRNN. We directly parameterize the weights as being low tensor rank $\mathbf{W} = \sum_{r=1}^R \mathbf{a}_r \otimes \mathbf{b}_r \otimes \mathbf{c}_r$ and perform gradient descent (more precisely, ADAM) on $\mathbf{a}_r, \mathbf{b}_r, \mathbf{c}_r$. For this, we use the framework of neural ODEs which allows backpropagating through solving an ODE [3].
>
> We believe that the addition of the training procedure pseudocode will help clarify this.
>
> **Q7. Truncated SVD**
>
> **A.** In line 168, we agree that this wording is unclear: we apply SVD and PARAFAC to the neural data itself rather than the weights. When we say SVD and PARAFAC are outperformed by ltrRNN we mean that ltrRNN has lower mean squared error in the test dataset than either SVD or PARAFAC models of the same rank. We included these as a baseline as it is very common in neuroscience to analyze low rank SVD or PARAFAC [4] models fit to neural activity.
>
> **Q8. Interpretable visualizations with ltrRNN**
>
> **A.** We first thank the reviewer for noting the typo on line 182; we will correct this in the revised paper. In Figure 3, we followed common practice [5] of convolving the spike times with a Gaussian kernel, with $\sigma=40$ ms (Supplementary Material B for details). However, note that in the Supplementary Material, we also illustrate an application of ltrRNN which is fitted to maximize the log-likelihood of the spike counts assuming a Poisson distribution (Supplementary Figure 5).
>
> There is a rich literature exploring the computational implications of chaotic dynamics in RNN models in neuroscience (e.g., [1]). However, the amount of variability in neural data (i.e., from trial to trial) precludes any definitive statement regarding whether fluctuations in recorded neural activity is due to the chaotic regime or noise. This question is especially fraught for RNNs due to technical issues in their training procedures that can prevent them from inferring chaotic dynamics from time series [6].
>
> **Q9. Uncovering low-rank structure by definition**
>
> **A.** The ltrRNN is indeed low-rank by definition. We thank the reviewer for pointing out that this sentence is misleading. In the revised paper we will rewrite this sentence as: "LtrRNNs enables inference of the tensor rank of the weights from the neural activity through crossvalidation."
>
>
> **Q10. Reference to adjoint-based automatic differentiation**
>
> **A.** We thank the reviewer for pointing us to this work which we were previously unaware of. We will incorporate the citation into the revised paper.
>
> **References**
>
> [1] Kadmon, Timcheck, and Ganguli. "Predictive coding in balanced neural networks with noise, chaos and delays." *NeurIPS* 2020.
>
> [2] Valente, Pillow, and Ostojic. "Extracting computational mechanisms from neural data using low-rank RNNs." *NeurIPS* 2022.
>
> [3] Chen, Rubanova, Bettencourt, and Duvenaud. "Neural ordinary differential equations." *NeurIPS* 2018.
>
> [4] Soulat, Keshavarzi, Margrie, and Sahani. "Probabilistic tensor decomposition of neural population spiking activity." *NeurIPS* 2022.
>
> [5] Park, Seth, Paiva, Li, and Principe. "Kernel methods on spike train space for neuroscience: a tutorial." *IEEE Signal Processing* 2013.
>
> [6] Mikhaeil, Monfared, and Durstewitz. "On the difficulty of learning chaotic dynamics with RNNs." *NeurIPS* 2022.

---

> > ### Comment · Reviewer_T6eW · 2023-08-12
> >
> > Thank you for your answers and clarifications for application specific jargon.
> > As the main weakness of the paper - the very application specific wording and method presentation, that was also pointed out by other reviewers - is addressed, my initial decision to accept the paper is reinforced.
> >  With the additional explanations I think this work is a compelling read for the NeurIPS community and helps bridging the gap between method and application for low-rank techniques.

---

### Official Review · Reviewer_X8YP · 2023-07-07

**Soundness:** 3 good
**Presentation:** 2 fair
**Contribution:** 3 good
**Rating:** 5
**Confidence:** 2

**Summary:**

In this paper, the authors explore the tensor rank of learning in artificial and biological neural networks. They showed that learning leads to low-tensor-rank weight updates, and derived upper bounds on the singular values of gradient dynamics of nonlinear RNNs, as well as on the matrix and tensor ranks in the linear case. Experimental results effectively support their model's conclusion.

**Strengths:**

1. Previous works have shown weight matrices in well-trained RNNs are low-rank. This paper focuses on whether tensors derived from weight matrices over the process of training are low-tensor-rank, which is meaningful and has a strong motivation.

2. The paper supports its results empirically.

3. The theoretical statements are detailed and valid.

**Weaknesses:**

1. From the perspective of researchers outside this field, a more structured presentation may be more conducive to understanding the contribution of work and promoting work.

**Questions:**

None.

**Limitations:**

Yes, the authors have adequately addressed the limitations and potential negative societal impact of their work.

---

> ### Author Rebuttal · Authors · 2023-08-10
>
> We thank the reviewer for noting that our paper has a *strong motivation*, and that our claims are supported both by our *empirical* and our *mathematical results*. We agree that providing a clear presentation of these results to researchers unfamiliar with neuroscience is important for the broader impact of our work. We briefly summarize the structural changes to the paper that we will make in response to all reviewers' feedback:
>
> * **Mathematical results.** As pointed out by other reviewers, the bounds we mathematically derive regarding the rank of the gradient of recurrent neural networks may be of interest to the broader machine learning community, but only come late in the paper. To address this, we will summarize them and their associated simulations earlier in the paper.
> * **Terminology.** Some terms such as "trial" in our neural data application are not standard in machine learning. To address this, we will define neuroscience-specific terminology more systematically.
> * **Main contributions.** Although the different sections of our paper cohesively support our main hypothesis that learning is low tensor rank, we aknowledge that a subset of them might be more of interest to any particular community at NeurIPS. To address this, we will add a main contributions section at the beginning of the paper highlighting its structure so that readers can navigate to the section that is most relevant to their research.
> * **LtrRNN pseudocode**. In light of multiple reviewers' comments and suggestions, we believe that understanding the fitting procedure of ltrRNN is key to understand the paper as a whole. To address this, we will i) include a short description of the ltrRNN fitting procedure in the main text ii) include pseudocode of ltrRNN fitting procedure.
> * **Supplementary material.** Many of our results rely on Supplementary Material (e.g. all theorems' proofs). To address this, we will more systematically reference specific sections of Supplementary Material.
>
> Overall, we believe that these structural changes will significantly improve the accessibility of our work. Nevertheless, any additional suggestions the reviewer may have to improve readability would be very welcome, as we want to make sure that our work is accessible to readers from across different backgrounds.

---

### Author Rebuttal · Authors · 2023-08-10

We thank the reviewers for their helpful and supportive comments. We are pleased to have received positive feedback regarding the novelty and interest of our submission from the reviewers, several of whom are self-described non-experts in neuroscience. We believe this highlights the potential for our work to be relevant to many research areas across the broader machine learning community.

We have made two substantial improvements to the paper in response to the reviewers' constructive criticisms. Results from our new analyses can be found in **Figures R1 and R2** of the associated 1-page PDF.

**New simulations.**
To investigate the generality of our results, we have now expanded our analyses by testing the low-tensor-rank learning hypothesis on three additional RNN models commonly used in neuroscience. These include:

* Sensory evidence accumulation task [1] where an RNN must learn to integrate a noisy instantaneous input (**Figure R1 i**).
* Contextual decision making task [2] where an RNN must decide which of two noisy inputs to integrate (**Figure R1 ii**).
* Working memory task [3] where an RNN must maintain a representation of a past input over time in order to compare it to a later input (**Figure R1 iii**).

Along with the motor adaptation example in the original paper, these four tasks span perceptual, motor and cognitive processes. Moreover, RNNs trained on these tasks have been proposed as models of different areas of the brain involved in those processes (e.g., [4,5]). Thus, these new experiments constitute a more systematic validation of the low-tensor-rank weights hypothesis we put forth.

We followed the same procedure to determine the tensor rank of learning for these tasks as for the task-trained RNN in our original submission (Figure  4). That is, we first trained an RNN on each of these tasks using gradient descent, then used PARAFAC to determine the tensor rank of the resulting neuron $\times$ neuron $\times$ iteration tensor of weights over learning. *In each of these tasks we found that the variance explained indeed saturated at low tensor ranks* (at $R=1,4,$ and $3$; **Figure R1 c**). We will add these results as a supplementary figure to the revised version of the paper.

**New data application.**
In addition, we applied the ltrRNN model to a new dataset consisting of neural recordings in mouse visual cortex during a perceptual learning task [6] (**Figure R2 a**). As a reminder, in the ltrRNN we fit an RNN to neural activity with the explicit constraint that the weight tensor has tensor rank $R$ (Figure 1; Supplementary Material A). We found that the performance of ltrRNN in this new dataset saturated at low ranks around $R = 3$ or $4$ (**Figure R2 b**). The fitted inputs distinguished rewarded from non-rewarded stimuli after the time at which the sensory stimulus was presented (**Figure R2 c**). Some of the trial factors appeared to track slow changes over learning while others remained relatively stable over trials (**Figure R2 d**). *These new results complement our original application to data from monkey motor cortical recordings during an adaptation task, demonstrating that the ltrRNN framework is not specific to a particular kind of learning, brain region, or species.*

***Overall we believe that our new simulations and data analyses, together with our mathematical results which give generic bounds on the ranks of gradient-based learning, validate the breadth of the ltrRNN framework.***

**Clarity and terminology.**
Several reviewers also noted that parts of the manuscript were difficult to understand due to (1) some details or definitions being hidden in the supplementary information and (2) undefined terminology (often, jargon specific to the neuroscience community). We strongly agree that it is important that our manuscript be accessible to the broader machine learning community, as we believe many of our results are of more general interest, beyond applications in neuroscience. To address this, we will make the following changes to the paper:

* To avoid overloading the reader, we had originally hidden some of the model specifications and training in the Supplementary Material. The reviewers fairly pointed out that this made it difficult to understand the paper. We will correct this by 1) incorporating many of these technical details back into the main text, and 2) by including pseudocode of the ltrRNN fitting procedure.
* To aid the non-specialist reader, we will add definitions of neuroscience-specific jargon.
* We will add a *main contributions* section to highlight the two principal components of our paper: 1) ltrRNN as a method for fitting neural data, and 2) the theoretical results regarding gradient learning dynamics. We believe this section will help readers from across different communities of NeurIPS better navigate the paper.

We have provided detailed responses to individual reviewers' questions below.

**References**

[1] Zoltowski, Pillown, and Linderman. "A general recurrent state space framework for modeling neural dynamics during decision-making." *ICML* 2020.

[2] Valente, Pillow, and Ostojic. "Extracting computational mechanisms from neural data using low-rank RNNs." *NeurIPS* 2022.

[3] Schuessler, Mastrogiuseppe, Dubreuil, Ostojic, and Barak. "The interplay between randomness and structure during learning in RNNs." *NeurIPS* 2020.

[4] Feulner, Perich, Chowdhury, Miller, Gallego, and Clopath. "Small, correlated changes in synaptic connectivity may facilitate rapid motor learning." *Nature Communications* 2022.

[5] Mante, Sussillo, Shenoy, and Newsome. "Context-dependent computation by recurrent dynamics in prefrontal cortex. *Nature* 2014.

[6] Khan, Poort, Chadwick, Blot, Sahani, Mrsic-Flogel, and Hofer. "Distinct learning-induced changes in stimulus selectivity and interactions of GABAergic interneuron classes in visual cortex." *Nature Neuroscience* 2018.

---

> ### Author Response · Authors · 2023-08-20
>
> As the author-reviewer discussion period is coming to an end, we would like to thank the reviewers for their overall positive response to our rebuttal. We are especially grateful to the reviewers for their suggestion to perform additional validations on simulations and new data, as we believe these have now significantly strengthened our paper.
>
> We remain available to answer any additional questions readers might have regarding the tensor rank of RNN gradients and its applications to task-trained RNNs or neural data analysis.

---

### Decision · Program_Chairs · 2023-09-21

**Decision:**

Accept (poster)

**Comment:**

The authors in this manuscript study the stages of learning in recurrent neural networks and how low-rank connectivity emerges. The authors use their observation that the low-dimensional space of the low-rank connectivity is fairly consistent over learning to improve learning algorithms. The theory of low-rank learning was followed up by experiments in the domain of motor learning. The reviewers agreed that the theoretical and practical aspects of the paper were interesting and well presented, and thus I recommend this work be accepted.